# VoiceBridge: Designing Latent Bridge Models for General Speech Restoration at Scale

## Abstract

Bridge models have recently been explored for speech enhancement tasks such as denoising, dereverberation, and super-resolution, while these efforts are typically confined to a single task or small-scale datasets, with constrained general speech restoration (GSR) capability at scale. In this work, we introduce VoiceBridge, a GSR system rooted in latent bridge models (LBMs), capable of reconstructing high-fidelity speech at fullband (*i.e.,* 48 kHz) from various distortions. By compressing speech waveform into continuous latent representations, VoiceBridge models the *diverse LQ-to-HQ tasks* (namely, low-quality to high-quality) in GSR with *a single latent-to-latent generative process* backed by a scalable transformer architecture. To better inherit the advantages of bridge models from the data domain to the latent space, we present an energy-preserving variational autoencoder, enhancing the alignment between the waveform and latent space over varying energy levels. Furthermore, to address the difficulty of HQ reconstruction from distinctively different LQ priors, we propose a joint neural prior, uniformly alleviating the reconstruction burden of LBM. At last, considering the key requirement of GSR systems, human perceptual quality, a perceptually aware fine-tuning stage is designed to mitigate the cascading mismatch in generation while improving perceptual alignment. Extensive validation across in-domain and out-of-domain tasks and datasets (*e.g.,* refining recent zero-shot speech and podcast generation results) demonstrates the superior performance of VoiceBridge. Demo samples can be visited at: `https://VoiceBridgedemo.github.io/`.

## 1 Introduction

Tractable Schrödinger Bridge (SB) models have proven to faithfully reconstruct the target distribution from an informative prior, overcoming the limitation of the noise prior in diffusion models (Chen et al., 2021; 2023; Li et al., 2025b). Recently, their advantages have been extended to speech enhancement tasks (Jukić et al., 2024; Wang et al., 2024a; Han et al., 2025), such as denoising (Jukić et al., 2024), dereverberation, and super-resolution (Li et al., 2025a), where the low-quality (LQ) observation naturally provides indicative information for the high-quality (HQ) target. Although these efforts have proposed innovations spanning forward process design (Jukić et al., 2024), model parameterization (Richter et al., 2025), training objectives (Zhang et al., 2025a), and cascaded architectures (Han et al., 2025; Li et al., 2025b), their designs are usually confined to a single task or small-scale datasets, not yet achieving general speech restoration (GSR) at scale to address various real-world speech degradations.

In this work, we propose *VoiceBridge*, a general speech restoration (GSR) system (Benesty et al., 2006; Kolbæk et al., 2016; Das et al., 2021) rooted in a latent bridge model (LBM), where the *diverse LQ-to-HQ tasks* in GSR are modeled with *a single latent-to-latent generative process* backed by a transformer architecture. By encoding both the LQ and HQ speech signals into continuous representations in a small latent space, VoiceBridge aims to preserve the advantages of the *data-to-data* sampling nature of bridge models from data space to latent space while achieving efficient training. Towards generalizable modeling, VoiceBridge explores combining the advantages of transformer architecture, namely good scalability that has been verified in both image (Peebles & Xie, 2023; Bao et al., 2023) and audio (Evans et al., 2024a) generation, with the bridge generative framework, further strengthening GSR performance at scale.

To better inherit the advantages of bridge generative models on speech enhancement tasks in the data domain, *e.g.*, bandwidth extension in the waveform domain (Li et al., 2025a), within LBMs, a natural idea is to preserve the characteristics in data space into latent representations. Motivated by this, we develop an *energy-preserving variational autoencoder (EP-VAE)* to achieve stronger consistency between the waveform and latent space than vanilla VAE (Kingma et al., 2019; Evans et al., 2024a). Specifically, we modify the VAE training objective, requiring a linear scaling operation in the latent space to be manifested in the decoded waveform space. In comparison with vanilla VAE that regularizes data reconstruction in a single scale, EP-VAE introduces reconstruction regularization over varying energy levels, naturally improving the waveform-latent consistency and resulting in a more structural latent space (Kouzelis et al., 2025; Yao et al., 2025; Skorokhodov et al., 2025; Yu et al., 2024) for LBM modeling.

Moreover, considering the difficulty of reconstructing the HQ target from distinctively different LQ priors caused by flexible degradation methods, such as noise, down-sampling, clipping, reverberation, vocal effects, and their random combinations (Zhang et al., 2025b; Serrà et al., 2022; Scheibler et al., 2024), we propose *a joint neural prior*, further enhancing the prior distribution of LBM. In detail, given the waveform encoder pre-trained by EP-VAE, we replace the reference signal from the LQ prior to the HQ target, enabling the encoder to uniformly reduce the distance between each LQ prior and the HQ target in the latent space. Similar to bridge models outperforming diffusion models when an informative prior is provided (Chen et al., 2023; Wang et al., 2024c; Li et al., 2025a), a joint neural prior enhances LBM-based GSR by further alleviating the *LQ-to-HQ* generation burden.

Given the joint neural prior and the HQ target encoded in a structural latent space, we have been able to establish a bridge transformer to tackle the various restoration tasks at scale. Considering the key requirement of GSR systems (Manjunath, 2009; Manocha et al., 2022; Babaev et al., 2024), human perceptual quality, which is a main difference from task-specific SE systems pursuing accurate waveform reconstruction (Li et al., 2025a; Jukić et al., 2024), we introduce a *perceptual-aware fine-tuning stage*. This design complements the LBM which is typically trained to fit the target distribution from prior distribution without explicitly considering human perception quality. Concretely, we introduce two popular speech metrics, PESQ (Rix et al., 2001) and UTMOS (Saeki et al., 2022), into the post-training stage of VoiceBridge, which optimizes both the LBM sampling and the EP-VAE decoding, minimizing their cascading mismatches in inference and aligning the entire generation process with human perceptual quality.

In summary, we make the following contributions.

- We design VoiceBridge, a LBM-based GSR system tackling *diverse LQ-to-HQ tasks* with *a single latent-to-latent generative process* backed a transformer architecture, and integrating multiple innovations to the LBM system.
- We propose *EP-VAE* to strengthen the consistency between the waveform and the latent spaces over varying energy levels, thereby inheriting the advantages of the bridge generative framework from the data domain into the latent space.
- We develop *joint neural prior* and *perceptual-aware fine-tuning*, further reducing the reconstruction burden of LBM while aligning it with human perceptual quality. We further conduct systematic studies on combining metric-based perceptual objectives with adversarial and feature-matching losses in the fine-tuning stage.
- Comprehensive validation across in-domain and out-of-domain tasks and datasets, including refining recent zero-shot speech and podcast generation results, shows the superiority of VoiceBridge in reconstructing high-fidelity 48 kHz speech from various distortions.

## 2 RELATED WORK

**Bridge-based speech enhancement.** Considering the advantages of the data-to-data generative framework on the tasks with an indicative prior, *e.g.*, speech denoising, dereverberation, and up-sampling, several recent works have explored the parameterization method (Richter et al., 2025), the noise schedule (Li et al., 2025a; Jukić et al., 2024), and the training objectives (Zhang et al., 2025a) for bridge-based speech enhancement models. However, their designs are usually proposed for a single task, including a recent work, AudioLBM (Li et al., 2025b), specialized in audio super-resolution. Moreover, a part of recent efforts are verified with narrow-band *e.g.*, 16 kHz or 24 kHz

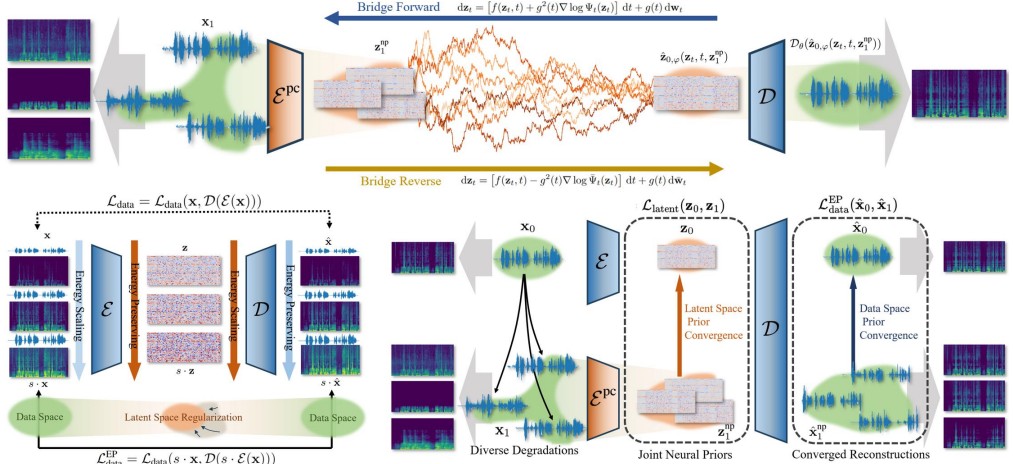

Figure 1: Overview of VoiceBridge. The upper part demonstrates the designed LBM-based GSR system. The lower part shows our approaches to building a structural latent space and converging a joint neural prior. On the left, *EP-VAE* requires alignment between the latent and data space at varying energy levels. On the right, *a joint neural* is encoded to reduce the distance between LQ priors and the HQ target, facilitating LBM reconstruction.

speech samples (Wang et al., 2025) or small-scale benchmark datasets (Jukić et al., 2024). These restrictions may limit the applicability of these designs to reconstruct 48 kHz high-fidelity speech samples from diverse real-world degradations (*i.e.*, GSR).

**General speech restoration.** Previous GSR systems have exploited mapping-based methods (Liu et al., 2022), adversarial training (Babaev et al., 2024), masked generative models (Li et al., 2024; Zhang et al., 2025b) or pretraining (Wang et al., 2025), conditional diffusion models (Serrà et al., 2022; Scheibler et al., 2024; Dhyani et al., 2025), and flow-matching methods (Niu, 2024; Liu et al., 2023a; Ku et al., 2025). In VoiceBridge, we design an LBM that extends the advantages of the *data-to-data* generative framework from specific SE tasks to GSR at scale and introduce three novel techniques that holistically improve generation results. We provide a more detailed introduction to related work in Appendix A.

## 3 VOICEBRIDGE

In this section, we first present our designed LBM-based GSR system, and then introduce the three innovations leading to holistic improvement, namely *EP-VAE*, *joint neural prior*, and *perceptual-aware generation*, respectively.

### 3.1 LATENT BRIDGE TRANSFORMER FOR GSR

In VoiceBridge, we construct the boundary distributions of bridge models, namely the high-quality target and the low-quality prior, as follows. Given a clean and full-band speech signal $\mathbf{x}_0 \in \mathbb{R}^L$, to simulate real-world speech distortions, we construct the degraded version $\mathbf{x}_1 \in \mathbb{R}^L$ with a degradation operator $\mathcal{T}$:

$$\mathbf{x}_1 = \mathcal{T}(\mathbf{x}_0), \tag{1}$$

where $\mathcal{T}$ is sampled from a range of speech degradation methods, including additive noise, reverberation, bandwidth limitation, clipping, vocal effects, and their random mixtures. Namely, for each ground-truth $\mathbf{x}_0$, VoiceBridge considers numerous different LQ prior $\mathbf{x}_1$, rather than considering a fixed degradation method as a single speech enhancement task (Jukić et al., 2024; Li et al., 2025a;b). A more detailed data construction process at the training stage is introduced in Appendix B.

Given constructed LQ and HQ waveform pairs $(\mathbf{x}_0, \mathbf{x}_1)$, we directly compress the speech waveform into latent representations, obtaining the latent target and prior, $\mathbf{z}_0 \in \mathbb{R}^{c \times l}$ and $\mathbf{z}_1 \in \mathbb{R}^{c \times l}$. Prior works (Li et al., 2025a; Jukić et al., 2024) mainly conduct bridge modeling in the data space, either in the waveform or spectrogram domain, to handle single tasks such as speech super-resolution or speech enhancement. However, we find that models in these domains struggle in the complex GSR scenario and 48kHz settings. Directly performing SB in the data space takes great modeling

and computational effort. Meanwhile, raw speech signals include massive redundant information at high frequencies, which can be effectively compressed by VAEs, resulting in a more compact latent space. Latent compression reduces sequence length by more than an order of magnitude, enabling a single 544M transformer LBM to handle all tasks. Details experiments proving the necessity of the *latent SB* design is shown in Appendix I

In comparison with alternative speech compression representations, the waveform latent naturally preserves the *data-to-data* generative process for the *LQ-to-HQ* speech tasks without suffering from area removal (Kong et al., 2025), which is important to a range of tasks in GSR. For example, for the down-sampling degradation and its mixture with other degradation methods, GSR systems are usually required to generate the 48 kHz waveform from its 2 kHz or even 1 kHz version (Andreev et al., 2023; Ku et al., 2025). In these settings, the prior information contained in the spectrograms (Mandel et al., 2023; Kong et al., 2025) and the latent space of the mel-spectrogram (Liu et al., 2022) will be very limited because of their large-scale area removal, which may hinder the final restoration performance as mentioned by (Lim et al., 2018; Kong et al., 2025).

Given these paired latents $(\mathbf{z}_0, \mathbf{z}_1)$, we model the generative process between them as a Tractable Schrödinger Bridge, a probabilistic interpolation between two marginal distributions over time. The SB problem seeks the stochastic trajectory $p_t$ connecting two distributions $p_0, p_1$ which minimizes the KL divergence to a reference diffusion process. When we take the marginal distributions $p_0, p_1$ to be Gaussian distributions around the latent pairs $(\mathbf{z}_0, \mathbf{z}_1)$, the generally intractable SB problem will have a closed-form solution (Bunne et al., 2023; Chen et al., 2023), where the interpolated distribution $p_t$ at time $t$ is a Gaussian:

$$p_t = \mathcal{N}(\frac{\alpha_t \bar{\sigma}_t^2}{\sigma_1^2} \mathbf{z}_0 + \frac{\bar{\alpha}_t \sigma_t^2}{\sigma_1^2} \mathbf{z}_1, \frac{\alpha_t^2 \bar{\sigma}_t^2 \sigma_t^2}{\sigma_1^2} \mathbf{I}) \tag{2}$$

where $\alpha_t, \sigma_t$ are defined by the reference diffusion process. A neural network is then applied to solve the bridge trajectory from any middle distribution $p_t$, eventually generating $\mathbf{z}_0 \sim p_0$ from given $\mathbf{z}_1 \sim p_1$ by sampling through the solved trajectory (Chen et al., 2023). Following (Chen et al., 2023; Li et al., 2025a), we re-parameterize the neural network to directly predict $\mathbf{z}_0$, optimizing:

$$\mathcal{L}_{\text{bridge}}(\varphi) = \mathbb{E}_{\substack{\mathbf{x}_0 \sim p_{\text{data}}, \ \mathbf{x}_1 = \mathcal{T}(\mathbf{x}_0) \\ \mathbf{z}_0 = \mathcal{E}(\mathbf{x}_0), \ \mathbf{z}_1 = \mathcal{E}(\mathbf{x}_1)}} \left\| \hat{\mathbf{z}}_{0,\varphi}(\mathbf{z}_t, t, \mathbf{z}_1) - \mathbf{z}_0 \right\|_2^2 \tag{3}$$

where $\hat{\mathbf{z}}_{0,\varphi}$ denotes the latent predicted by the neural network with trainable parameters $\varphi$; $\mathbf{z}_0, \mathbf{z}_1$ are the paired latent representations encoded from the HQ signal and its degraded LQ versions, respectively; $\mathbf{z}_t$ is constructed with Equation 2. For generalizable training, we explore combining the advantage of transformer architecture, namely good scalability as shown in diffusion-based both image (Peebles & Xie, 2023; Bao et al., 2023) and audio generation (Evans et al., 2024b), with the bridge generative framework. More discussion of tractable Schrödinger bridge models is provided in Appendix C. The details of network architecture are introduced in Appendix D.

## 3.2 ENERGY-PRESERVING VAE

As shown in Equation 3, the prior and target in VoiceBridge, $(\mathbf{z}_0, \mathbf{z}_1)$, are compressed from the corresponding waveform $(\mathbf{x}_0, \mathbf{x}_1)$. The compression network, namely VAE (Kingma et al., 2019), is employed to reduce the redundant information in waveform space to allow efficient training, while ensuring the reconstruction quality. Generally, the training of audio waveform VAE (Evans et al., 2024b) with network parameters $\theta$ optimizes the objectives on data-space reconstruction $\mathcal{L}_{\text{data}}$ and latent-space regularization $\mathcal{L}_{\text{latent}}$:

$$\mathcal{L}_{\text{vae}} = \mathcal{L}_{\text{data}}(\mathcal{D}_\theta(\mathcal{E}_\theta(\mathbf{x})), \mathbf{x}) + \mathcal{L}_{\text{latent}}(\mathcal{E}_\theta(\mathbf{x}), \mathbf{z}_{\text{ref}}), \tag{4}$$

where the term $\mathcal{L}_{\text{data}}$ is typically composed of reconstruction loss, adversarial loss, and feature matching loss between input signal $\mathbf{x}$ and reconstructed version $\mathcal{D}(\mathcal{E}(\mathbf{x}))$, while the term $\mathcal{L}_{\text{latent}}$ represents the KL divergence regularization of the latent $\mathbf{z} = \mathcal{E}(\mathbf{x})$ with a known prior distribution $\mathbf{z}_{\text{ref}}$, *e.g.*, standard Gaussian. In recent task-specific bridge-based speech enhancement works, the *data-to-data* generative framework effectively exploits the instructive information contained in the LQ observation in either waveform space (Li et al., 2025a) or STFT representations (Jukić et al., 2024; Kong et al., 2025; Han et al., 2025). Hence, when designing a *latent-to-latent* GSR system, we naturally explore preserving the advantages of bridge models from the data space to the encoded

latent space. In VoiceBridge, to facilitate latent SB modeling, we introduce a novel technique, EP-VAE, to enhance the consistency between data and latent from the perspective of scale equivariance. Specifically, we add an EP constraint into the VAE training objective, expanding the data-space loss $\mathcal{L}_{\text{data}}$ in Equation 4 to continuous energy levels with $\mathcal{L}_{\text{data}}^{\text{ep}}$:

$$\mathcal{L}_{\text{ep-vae}} = \mathcal{L}_{\text{data}}^{\text{ep}}(\mathcal{D}_\theta(s \cdot \mathcal{E}_\theta(\mathbf{x})), s \cdot \mathbf{x}) + \mathcal{L}_{\text{latent}}(\mathcal{E}_\theta(\mathbf{x}), \mathbf{z}_{\text{ref}}), \tag{5}$$

where $s \sim \mathcal{U}(0.5, 2)$ is a random scaling factor sampled from a uniform distribution. The EP-VAE is required to maintain the rescaling of latent $\mathbf{z}$ on the reconstructed signal $\mathbf{x}$: when the latent energy is rescaled by $s$ times to $s \cdot \mathbf{z} = s \cdot \mathcal{E}(\mathbf{x})$, the reconstruction signal should have energy rescaled by $s$ times to $s \cdot \mathbf{x}$ as well. By strengthening waveform-latent alignment with extra regularization at diverse energy levels, we obtain a more structural latent space, which facilitates LBM to reconstruct the distribution of latent HQ target, namely $\mathbf{z}_0$, from the informative latent prior $\mathbf{z}_1$, boosting the performance for high-quality GSR.

### 3.3 PRIOR CONVERGENCE

In GSR systems, task diversity is one of the key features. Namely, one target $\mathbf{x}_0$ can suffer from diverse degradations as the generation prior $\mathbf{x}_1$, and therefore a GSR system should be able to reconstruct the HQ target from a flexible LQ prior rather than a fixed version. However, in the latent space, when the LQ priors $\mathbf{z}_1 = \mathcal{E}(\mathbf{x}_1)$ are distinctly different from each other as shown in Figure 2, it inevitably increases the difficulty for an LBM to model *diverse LQ-to-HQ* tasks with a *single latent-to-latent* generative process.

Hence, we present *joint neural prior*, aiming at uniformly reducing the distance between each LQ prior $\mathbf{z}_1$ and the ground-truth target $\mathbf{z}_0$, thereby facilitating the bridge generation process. Specifically, given the pre-trained EP-VAE encoder $\mathcal{E}$ used for $\mathbf{x}_1$, we fine-tune it to obtain a new version, $\mathcal{E}^{\text{np}}$, which is capable of converging LQ priors $\mathbf{z}_1$ to a joint neural prior, $\mathbf{z}_1^{\text{np}}$. Given the waveform pair $(\mathbf{x}_0, \mathbf{x}_1)$, we encode $\mathbf{x}_0$ with the pre-trained $\mathcal{E}$, and $\mathbf{x}_1$ with a trainable encoder $\mathcal{E}^{\text{np}}$ initialized from $\mathcal{E}$. Their latent representations $\mathbf{z}_0 = \mathcal{E}(\mathbf{x}_0)$ and $\mathbf{z}_1^{\text{np}} = \mathcal{E}^{\text{np}}(\mathbf{x}_1)$ are then decoded via the shared pre-trained EP-VAE decoder $\mathcal{D}$ into waveform

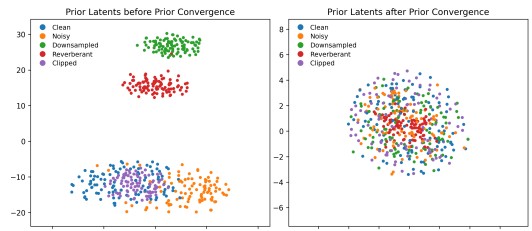

Figure 2: The tSNE visualization (Maaten & Hinton, 2008) of the prior latent before and after *prior convergence* from augmented VCTK (Veaux & Yamagishi, 2017). Note the difference between the scale of axes of the two figures.

$\hat{\mathbf{x}}_0 = \mathcal{D}(\mathbf{z}_0)$ and $\hat{\mathbf{x}}_1^{\text{np}} = \mathcal{D}(\mathbf{z}_1^{\text{np}})$. Then, we reduce the distance between each LQ prior and the ground-truth target from both data and latent spaces with:

$$\mathcal{L}_{\text{np-enc}} = \mathcal{L}_{\text{data}}^{\text{ep}}(\mathcal{D}(s \cdot \mathcal{E}_\theta^{\text{np}}(\mathbf{x}_1)), s \cdot \hat{\mathbf{x}}_0) + \mathcal{L}_{\text{latent}}(\mathcal{E}_\theta^{\text{np}}(\mathbf{x}_1), \mathbf{z}_0), \tag{6}$$

where $\mathcal{L}_{\text{latent}}$ has been the joint neural prior loss consisting of MSE loss and cosine similarity loss as introduced in Appendix D, and $\mathcal{L}_{\text{data}}^{\text{ep}}$ is the data-space loss in EP-VAE training modified for prior convergence. Prior works (Fang et al., 2021) applies a KL convergence loss to a VAE for speech denoising. In contrast, we add objectives including cosine similarity in latent space and multiple supervision signals in the reconstructed data space, in order to reduce the distance between the latents in all geometrical measures. We prove our method achieves better results in Appendix I.

Namely, as shown in Equation 6, the encoder $\mathcal{E}^{\text{np}}$ is fine-tuned with changed objectives to converge different LQ priors to HQ target, rather than still learning the latent presentation for reconstruction as Equation 5. Moreover, the EP constraint is preserved. When reducing the distance between LQ priors $\mathbf{z}_1$ and the HQ target $\mathbf{z}_0$ to achieve $\mathbf{z}_1^{\text{np}}$, scale equivariance is simultaneously required, which inherits the EP property to measure the consistency at varying energy levels. As shown in Figure 2, this technique significantly closes the distance between different LQ priors and the HQ target in the latent space, and naturally reduces the burden of LBM modeling. A further quantitative analysis on the prior distance is shown in Appendix G

### 3.4 PERCEPTUAL-AWARE GENERATION

After training the EP-VAE encoder $\mathcal{E}$ with Equation 5 for encoding HQ target, its fine-tuned version $\mathcal{E}^{\text{np}}$ with Equation 6 for encoding diverse LQ priors, and the shared decoder $\mathcal{D}$, we can already

Table 1: Evaluation results on three GSR testsets, including a standard benchmark, a simulated testset whose training split is outside our training dataset for cross evaluation, and an in-the-wild dataset. Best is bolded, and second best is underlined. * denotes results taken from the original paper. VoiceBridge outperforms most baselines in nearly all metrics, demonstrating strong restoration ability in complex environments.

| Voicefixer-GSR (Liu et al., 2022) | | | | | | | |
|---|---|---|---|---|---|---|---|
| Model | PESQ (↑) | SIG (↑) | BAK (↑) | OVRL (↑) | UTMOS (↑) | WV-MOS (↑) | NISQA (↑) |
| Input | 1.943 | 2.962 | 2.858 | 2.395 | 2.633 | 1.965 | 2.674 |
| Miipher | 1.491 | 3.335 | 3.992 | 3.051 | 3.617 | 4.171 | 4.136 |
| VoiceFixer | 2.043 | 3.302 | 3.971 | 3.005 | 3.439 | 3.800 | 4.159 |
| Resemble-Enhance | 2.000 | 3.412 | 4.045 | 3.137 | 3.608 | 4.224 | **4.507** |
| UniverSE++ | 2.346 | 3.275 | 3.961 | 2.976 | 3.376 | 3.648 | 4.009 |
| AnyEnhance | - | 3.406* | 4.073* | 3.136* | - | - | 4.308* |
| **VoiceBridge** | **2.436** | **3.469** | **4.083** | **3.210** | **4.330** | **4.246** | 4.098 |

| DNS-with-Reverb (Reddy et al., 2021a) | | | | | | | |
|---|---|---|---|---|---|---|---|
| Model | PESQ (↑) | SIG (↑) | BAK (↑) | OVRL (↑) | UTMOS (↑) | WV-MOS (↑) | NISQA (↑) |
| Input | 1.176 | 1.767 | 1.481 | 1.377 | 1.323 | $-0.019$ | 1.689 |
| Miipher | 1.324 | 3.401 | 3.975 | 3.128 | 2.973 | 3.195 | 3.629 |
| VoiceFixer | 1.407 | 3.355 | 3.982 | 3.052 | 2.526 | 2.961 | 3.704 |
| Resemble-Enhance | 1.154 | 3.502 | 4.021 | 3.205 | 2.756 | 3.248 | **4.583** |
| UniverSE++ | 1.098 | 2.548 | 3.649 | 2.244 | 1.669 | 2.463 | 3.006 |
| AnyEnhance | - | 3.500* | 4.040* | 3.204* | - | - | 3.722* |
| **VoiceBridge** | **1.521** | **3.552** | **4.045** | **3.270** | **3.373** | **4.188** | 4.563 |

| DNS-Real (Reddy et al., 2021a) | | | | | | | |
|---|---|---|---|---|---|---|---|
| Model | PESQ (↑) | SIG (↑) | BAK (↑) | OVRL (↑) | UTMOS (↑) | WV-MOS (↑) | NISQA (↑) |
| Input | - | 2.985 | 2.510 | 2.212 | 1.940 | 1.483 | 2.160 |
| Miipher | - | 3.325 | 3.976 | **3.171** | 3.911 | 3.171 | 4.124 |
| VoiceFixer | - | 3.174 | 3.919 | 2.875 | 2.351 | 2.938 | 3.529 |
| Resemble-Enhance | - | 3.395 | **3.993** | 3.100 | 2.767 | 3.298 | 4.329 |
| UniverSE++ | - | 2.999 | 3.660 | 2.641 | 2.306 | 2.603 | 3.317 |
| AnyEnhance | - | **3.488*** | 3.977* | 3.161* | - | - | - |
| **VoiceBridge** | - | 3.447 | 3.985 | 3.166 | **4.197** | **4.034** | **4.497** |

create a structured latent space where the different LQ priors have converged. Then, we train the bridge transformer model that maps the joint neural prior and the HQ target using Equation 3, while the LQ prior and condition $\mathbf{z}_1$ have been changed to $\mathbf{z}_1^{\mathrm{np}}$ encoded by $\mathcal{E}^{\mathrm{np}}$, and the HQ target $\mathbf{z}_0$ is compressed from $\mathbf{x}_0$ with EP-VAE encoder $\mathcal{E}$.

Furthermore, different from previous bridge models designed for a specific SE task that pursue target waveform reconstruction accuracy, GSR systems are required to generate speech samples with high perceptual quality (Manjunath, 2009; Manocha et al., 2022; Babaev et al., 2024). This key feature motivates us to integrate the human perceptual quality into the entire generation process of VoiceBridge, further enhancing LBM-based GSR quality. Specifically, we perform a post-training stage for both the LBM and the EP-VAE decoder, mitigating their cascading mismatches in inference as well as aligning both the bridge sampling and VAE decoding with human perceptual quality. The post-training objectives for LBMs and the decoder include:

$$\mathcal{L}_{\mathrm{pa\text{-}gen}} = \mathcal{L}_{\mathrm{bridge}}(\varphi) + \mathcal{L}_{\mathrm{data}}(\mathcal{D}_\theta(\hat{\mathbf{z}}_{0,\varphi}(\mathbf{z}_t, t, \mathbf{z}_1^{\mathrm{np}})), \mathbf{x}_0) + \mathcal{L}_{\mathrm{perc}}(\mathcal{D}_\theta(\hat{\mathbf{z}}_{0,\varphi}(\mathbf{z}_t, t, \mathbf{z}_1^{\mathrm{np}})), \mathbf{x}_0), \quad (7)$$

where $\mathcal{L}_{\mathrm{perc}}$ denotes a PESQ loss and a UTMOS loss, jointly align the bridge sampling and the decoding processes well with subjective quality judgments. Intuitively, such a post-training procedure helps the bridge model to generate the speech samples on the sub-manifold of the latent space corresponding to the waveform with high perceptual quality, as well as encourages the decoder to generate HQ audio. As the encoder is frozen in this training section, this technique does not affect the good latent properties established by EP-VAE and prior convergence, allowing the utilization of each proposed innovation in VoiceBridge.

### 3.5 TRAINING PIPELINE OVERVIEW

In actual training, VoiceBridge integrates EP-VAE and the joint neural prior, and achieves perceptual-aware generation through a four-stage training pipeline: (1) pre-training an EP-VAE on

Table 3: Evaluation results on three denoising benchmarks. Best is bolded and second best is underlined. VoiceBridge achieves strong performance on the traditional speech enhancement task compared with both other GSR models and specific SE models.

| VB-Demand Valentini-Botinhao (2017) | | | | | | | |
|---|---|---|---|---|---|---|---|
| Model | PESQ (↑) | SIG (↑) | BAK (↑) | OVRL (↑) | UTMOS (↑) | WV-MOS (↑) | NISQA (↑) |
| Input | 1.972 | 3.343 | 3.124 | 2.694 | 3.063 | 2.985 | 2.998 |
| SBSE | 2.140 | 2.571 | 3.658 | 2.946 | 3.407 | 3.819 | 3.976 |
| VoiceFixer | 2.454 | 3.454 | 4.047 | 3.174 | 3.645 | 4.129 | 4.433 |
| Resemble-Enhance | 2.352 | 3.454 | 3.986 | 3.141 | 3.678 | 4.270 | 4.503 |
| UniverSE++ | **3.020** | **3.486** | 4.042 | 3.198 | 3.968 | 4.385 | 4.503 |
| **VoiceBridge** | 2.819 | 3.471 | **4.055** | **3.199** | **4.288** | **4.387** | **4.513** |

| WSJ0-CHiME3 Barker et al. (2015) | | | | | | | |
|---|---|---|---|---|---|---|---|
| Model | PESQ (↑) | SIG (↑) | BAK (↑) | OVRL (↑) | UTMOS (↑) | WV-MOS (↑) | NISQA (↑) |
| Input | 1.248 | 2.486 | 1.868 | 1.795 | 1.823 | 0.128 | 1.520 |
| SGMSE+ | 1.723 | 3.517 | 3.946 | 3.191 | 3.282 | 3.547 | 4.294 |
| StoRM | **1.826** | 3.452 | **4.115** | 3.214 | 3.468 | 3.618 | 4.611 |
| VoiceFixer | 1.493 | 3.293 | 4.059 | 3.052 | 2.863 | 3.255 | 3.921 |
| Resemble-Enhance | 1.171 | 3.524 | 4.113 | 3.263 | 2.749 | 3.474 | 4.610 |
| UniverSE++ | 1.320 | 3.173 | 3.997 | 2.914 | 2.984 | 3.248 | 3.918 |
| **VoiceBridge** | 1.727 | **3.536** | 4.093 | **3.280** | **4.440** | **4.356** | **4.632** |

| VCTK + DEMAND with SNR=-15dB | | | | | | | |
|---|---|---|---|---|---|---|---|
| Model | PESQ (↑) | SIG (↑) | BAK (↑) | OVRL (↑) | UTMOS (↑) | WV-MOS (↑) | NISQA (↑) |
| Input | 1.069 | 2.091 | 1.690 | 1.453 | 1.352 | -1.689 | 1.129 |
| Voicefixer | 1.225 | 2.558 | 3.280 | 2.313 | 2.423 | 2.278 | 2.848 |
| Resemble-Enhance | 1.168 | **3.194** | 3.396 | 2.640 | 3.394 | 2.440 | 3.130 |
| UniverSE++ | 1.211 | 2.780 | 3.202 | 2.374 | 2.950 | 2.772 | 2.271 |
| **VoiceBridge** | **1.352** | 3.039 | **3.690** | **2.698** | **3.479** | **2.755** | **3.270** |

clean speech with Equation 4; (2) fine-tuning the EP-VAE encoder to learn a joint neural prior on distorted inputs with Equation 6; (3) training the transformer model to solve the SB in latent space with Equation 3 and (4) jointly fine-tuning the LBM and the decoder to enhance perceptual alignment with Equation 7.

## 4 EXPERIMENTS

### 4.1 EXPERIMENTAL SETUP

We train and evaluate VoiceBridge on GSR task and test its zero-shot generalization to downstream tasks, especially out-of-domain (OOD) tasks. Our

Table 2: MOS results on VoiceFixer-GSR and DNS-Real datasets, covering both simulated and real data. VoiceBridge outperforms other baselines on subjective evaluation results collected in human listening experiments.

| | VF | RE | USE | VB | GT |
|---|---|---|---|---|---|
| Voicefixer-GSR | 3.77 | 3.97 | 2.91 | **4.32** | 4.43 |
| DNS-Real | 2.90 | 3.02 | 2.04 | **4.28** | - |

full training corpus contains approximately 1138 hours of clean speech audio at 48 kHz, constructed by combining multiple public datasets. Distortions are synthesized using publicly available noise datasets and room impulse responses, following previous works (Liu et al., 2022; Zhang et al., 2025b). The compression network is a 156M-parameter Oobleck VAE (Evans et al., 2024b). The LBM is built with a 544M-parameter Transformer backbone (Evans et al., 2024a). The training process is described in Section 3.5. We provided detailed information on the VAE compression network, datasets, inference process, and evaluation methods in Appendix D, E, H, and F, respectively.

### 4.2 GENERAL SPEECH RESTORATION

To comprehensively assess our GSR performance at scale, we first evaluate it on three benchmark datasets following previous works: VoiceFixer-GSR (Liu et al., 2022), DNS-with-Reverb (Reddy et al., 2021a), and DNS-Real-Data (Reddy et al., 2021a). Notably, data from the DNS-Challenge (Reddy et al., 2021a) are excluded in our training data, providing a mismatched training-testing condition for VoiceBridge. We compare VoiceBridge against other GSR models including Miipher (Koizumi et al., 2023), VoiceFixer (Liu et al., 2022), Resemble-Enhance (Niu, 2024), UniverSE++ (Scheibler et al., 2024) and AnyEnhance (Zhang et al., 2025b). Despite these models,

we further compare VoiceBridge to close-sourced models with there reported results or using there demo samples in Appendix F.

VoiceBridge and the four baselines have included five classes of generative models: bridge, mapping-based network, flow-matching, diffusion, and masked generative models. Such diversity enables a broad comparison across generative modeling strategies. Recently, non-intrusive metrics which measure perceptual quality without reference signals are believed to be more relevant to generative speech restoration (Manjunath, 2009; Manocha et al., 2022; Babaev et al., 2024). In our evaluation, we report both intrusive and non-intrusive metrics, incorporating PESQ (Rix et al., 2001), DNSMOS (Reddy et al., 2021b) (SIG, BAK, OVRL), UTMOS (Saeki et al., 2022), WV-MOS (Ogun et al., 2023), and NISQA (Mittag et al., 2021).

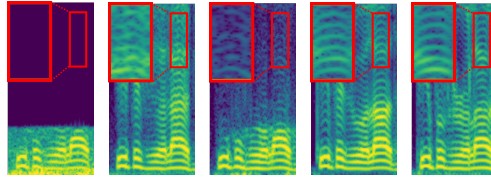

(a) LQ  (b) VF  (c) RE  (d) VB  (e) GT

Figure 3: STFT spectrograms of the same piece of speech restored by different models (a) Low-Quality Signal. (b) VoiceFixer (b) Resemble-Enhance (d) VoiceBridge (Ours) (e) Ground Truth.

The GSR evaluation results are shown in Table 1. VoiceBridge achieves the best or the second-best results on almost every metric across all datasets, demonstrating strong multi-degradation restoration capability on both simulated and real-world audios. We further conduct a user study with the publicly available VoiceFixer-GSR and DNS-Real to investigate human preference for each GSR system. The result, as shown in Table 2, demonstrates VoiceBridge's superior performance, achieving significantly higher Mean Opinion Score (MOS) on both simulated and real audio samples. A case study is shown in Figure 3, where VoiceBridge faithfully generates different frequency components of the ground truth in comparison with other methods. All demonstrated results of VoiceBridge are achieved with 4 latent inference steps, ensuring computational efficiency. We provide more details of inference efficiency in Appendix H.

### 4.3 ADDITIONAL RESULTS

As VoiceBridge is built with a scalable architecture and aims to achieve high-fidelity GSR at scale, we further evaluate its zero-shot performance on more tasks, and especially, the tasks with *out-of-domain* (OOD) degradation levels and types. We tested Voice-

Table 4: Codec Artifact Removal on Encodec-reconstructed audio from the VCTK testset. Though never trained on this task, VoiceBridge exhibits strong zero-shot performance in enhancing speech quality.

| VCTK Veaux & Yamagishi (2017) reconstructed by Encodec Défossez et al. (2022) | | | | |
|---|---|---|---|---|
| Method | PESQ(↑) | WVMOS(↑) | UTMOS(↑) | DNSMOS(↑) | NISQA(↑) |
| Input | 2.210 | 4.036 | 2.889 | 2.822 | 3.661 |
| Voicefixer | 1.984 | 3.956 | 3.229 | 3.003 | 4.436 |
| Resemble-Enhance | 2.059 | 4.054 | 3.294 | 3.063 | 4.665 |
| UniverSE++ | 2.320 | 4.002 | 2.954 | 2.868 | 3.861 |
| **VoiceBridge** | **2.341** | **4.109** | **3.675** | **3.109** | 4.452 |

Bridge on the task of traditional Speech Enhancement (SE), comparing it to task-specific models including SGMSE+ (Richter et al., 2023; Welker et al., 2022), StoRM (Lemercier et al., 2023b) and SBSE (Jukić et al., 2024) on benchmarks involving VoiceBank-DEMAND (Valentini-Botinhao, 2017) and WSJ-CHiME (Barker et al., 2015). We additionally curate a VCTK (Veaux & Yamagishi, 2017) + DEMAND (Thiemann et al., 2013) dataset, with a setting at $-15\,\mathrm{dB}$ SNR, which is well beyond our training settings, to assess the robustness of VoiceBridge under extreme noise conditions unseen in training. Apart from SE, evaluation for other tasks has also been conducted for bandwidth extension and dereverberation, with details shown in Appendix F. Moreover, as a probabilistic generative model, VoiceBridge captures target distributional information rather than establishing a point-to-point mapping function. Therefore, it shows strong zero-shot ability on OOD restoration tasks that are unseen during the training stage. We evaluate VoiceBridge and other GSR models on codec artifact removal, where all models are tested to repair reconstructed audio from VCTK by the Encodec (Défossez et al., 2022) model operating at 3 kbps bitrate.

The evaluation results of denoising are shown in Table 3, where VoiceBridge outperforms dataspace SB models SBSE (Jukić et al., 2024), diffusion models, and flow-matching models on various metrics. Note that the WSJ0 dataset, which the SGMSE+ and StoRM models are trained on, is excluded from our training dataset, exceptionally demonstrating the superiority of VoiceBridge. For the codec artifact removal results, as shown in Table 4, VoiceBridge consistently outperforms

Table 5: Enhancement results on generated speech by MoonCast and MaskGCT on the Seed-TTS benchmark. VoiceBridge exhibits consistent improvement on generated speech quality.

| Seed-TTS-Eval Anastassiou et al. (2024) with MoonCast Ju et al. (2025) | | | | | | | |
|---|---|---|---|---|---|---|---|
| Method | WVMOS(↑) | UTMOS(↑) | SIG(↑) | BAK(↑) | OVRL(↑) | NISQA(↑) | WER(↓) |
| Input | 3.971 | 3.623 | 3.451 | 3.880 | 3.120 | 3.922 | 5.93% |
| Voicefixer | 3.832 | 3.481 | 3.433 | 3.993 | 3.145 | 4.004 | 5.51% |
| Resemble-Enhance | 3.612 | 3.291 | 3.424 | 3.972 | 3.176 | 4.121 | 12.88% |
| UniverSE++ | 3.747 | 3.446 | 3.459 | 3.957 | 3.143 | 4.217 | 8.59% |
| VoiceBridge | **4.077** | **3.807** | **3.482** | **4.019** | **3.208** | **4.443** | **5.40%** |

| Seed-TTS-Eval Anastassiou et al. (2024) with MaskGCT Wang et al. (2024b) | | | | | | | |
|---|---|---|---|---|---|---|---|
| Method | WVMOS(↑) | UTMOS(↑) | SIG(↑) | BAK(↑) | OVRL(↑) | NISQA(↑) | WER(↓) |
| Input | 3.673 | 3.427 | 3.529 | 3.793 | 3.122 | 3.964 | **5.63%** |
| Voicefixer | 3.483 | 3.288 | 3.478 | 4.012 | 3.186 | 4.349 | 6.02% |
| Resemble-Enhance | 3.818 | 3.594 | **3.580** | 4.071 | **3.312** | 4.691 | 7.78% |
| UniverSE++ | 3.324 | 3.277 | 3.475 | 4.016 | 3.180 | 4.308 | 7.21% |
| VoiceBridge | **4.156** | **4.040** | 3.565 | **4.088** | 3.311 | **4.764** | 5.69% |

Figure 4: Ablation study with different GSR metrics. The horizontal axis shows training steps, and the vertical axis displays performance. The models differ in whether *EP-VAE* and *joint neural prior* are employed.

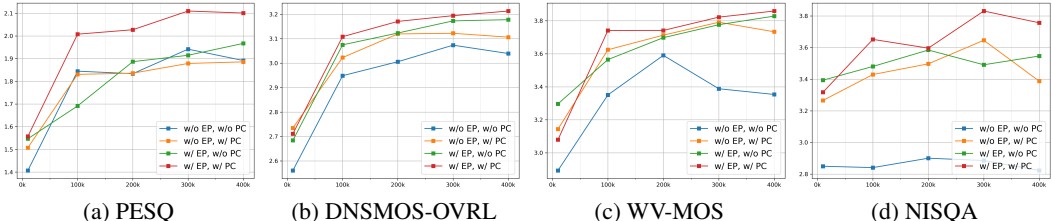

(a) PESQ  (b) DNSMOS-OVRL  (c) WV-MOS  (d) NISQA

baseline models on various metrics, exhibiting the zero-shot performance of VoiceBridge on this OOD task.

## 4.4 QUALITY IMPROVEMENT FOR SPEECH GENERATION

Beyond codec artifact removal, we test another application scenario of GSR models, refining the generation results of recent speech and podcast generation methods, namely, further enhancing the perceptual quality of text-to-speech (TTS) models. For validation, we take two recent methods, MaskGCT (Wang et al., 2024b) and MoonCast (Ju et al., 2025), as the TTS baseline methods to synthesize speech on the Seed-TTS (Boson AI, 2025; Peng et al., 2025) benchmark, and then apply VoiceBridge and other GSR baselines to refine the generated speech samples. We evaluate using both GSR and TTS metrics. As shown in Table 5, VoiceBridge reliably improves the quality of recent speech generation methods in a zero-shot manner. Importantly, other GSR models may fail to achieve high-fidelity GSR at scale, and therefore they are not able to reliably refine recent TTS methods, while VoiceBridge consistently refines these methods.

## 4.5 ABLATION STUDIES

We conduct an ablation study on our designs by training four model variants: with or without EP, and with or without the joint neural prior. All variants share the same latent bridge and decoder training pipeline for fair comparison. As shown in Figure 4, the baseline model without EP and joint neural prior consistently underperforms the other variants. Introducing the joint neural prior or adding the EP constraint leads to a noticeable improvement. The model with both techniques achieves the best overall performance, demonstrating the complementary benefits of these two mechanisms.

Further, we conduct another ablation study on the training objective and strategy during the perceptual aware fine-tuning stage. Based on the pre-trained LBM, we further conducted 9 groups of experiments with different settings in the post-training stage, to study the effect of different training objective and whether to adjust the decoder in this stage. The experiments are divided into 2 main groups. In the first group, we keep the decoder frozen, and use the data-space loss between decoded audio and clean audio to fine-tune the bridge model. In the second group, the decoder and bridge are jointly fine-tuned together.

Table 6: Evaluation results on VoiceFixer-GSR (Liu et al., 2022) for different perceptual-aware fine-tuning processes of VoiceBridge.

| Finetuned Module | Finetuning Objective | PESQ (↑) | DNSMOS (↑) | UTMOS (↑) | WV-MOS (↑) | NISQA (↑) |
|---|---|---|---|---|---|---|
| | | | Voicefixer-GSR Liu et al. (2022) | | | |
| Bridge | $\mathcal{L}_{lat.}$ | 2.067 | 3.031 | 3.645 | 4.153 | 4.067 |
| | $+\mathcal{L}_{rec}$ | 2.125 | 3.049 | 3.649 | 4.207 | 3.524 |
| | $+\mathcal{L}_{rec} + \mathcal{L}_{gan}$ | 2.058 | 3.069 | 3.533 | 4.055 | 3.362 |
| | $+\mathcal{L}_{rec} + \mathcal{L}_{hf}$ | 2.171 | 3.038 | 3.625 | 4.122 | 3.466 |
| | $+\mathcal{L}_{rec} + \mathcal{L}_{gan} + \mathcal{L}_{hf}$ | 2.182 | 3.049 | 3.599 | 4.158 | 3.487 |
| Bridge + Decoder | $+\mathcal{L}_{rec}$ | 2.297 | 2.952 | 3.588 | 3.972 | 2.301 |
| | $+\mathcal{L}_{rec} + \mathcal{L}_{gan}$ | 2.169 | 2.938 | 3.475 | 3.709 | 3.091 |
| | $+\mathcal{L}_{rec} + \mathcal{L}_{hf}$ | **3.014** | 3.025 | **4.267** | 4.252 | 3.167 |
| | $+\mathcal{L}_{rec} + \mathcal{L}_{gan} + \mathcal{L}_{hf}$ | 2.722 | **3.091** | 4.228 | **4.366** | **4.172** |

We test the following set of loss combinations, shown in Table 6. Firstly, a setting for continually training with the original objective (latent space MSE loss) $\mathcal{L}_{latent}$ is provided as baseline reference (Line 1). For the other settings, we include the MRSTFT reconstruction loss $\mathcal{L}_{rec}$, which is used for VAE training, as the basic data-space loss for the post-training stage (Line 2,6). Then, we add the GAN loss $\mathcal{L}_{gan}$, which is composed of the adversarial and feature matching loss and the perceptual term $\mathcal{L}_{perc}$ to the training objectives, first separately (Line 3,4,7,8), then together (Line 5,9). All these variants are applied to both the two groups in which the decoder is frozen or jointly finetuned. All hyper-parameters are aligned with the VAE training stage in Appendix D.

As shown in Table 6, conducting post-training with the decoder frozen makes little improvement to the restoration system, regardless of the training objective we use. Within the group which we do joint post-training of both the bridge and decoder, adding only the reconstruction loss or with the GAN loss for the decoder also has negligible or even negative effects. Interestingly, adding the perceptual term $\mathcal{L}_{perc}$ has a significant impact on the corresponding metrics, PESQ and UTMOS, but causes a decrease in other metrics. As pointed out in PESQetarian (de Oliveira et al.), including such metrics as training objectives leads to severe overfitting and degrades model output. In fact, in this group, the generated audio in this group shares a common artifact, which severely degrades the listening experience. This phenomenon can be comprehended as that, because the scoring network or algorithm for metric evaluation is frozen during training, the decoder learns to attack the evaluation network or algorithm, generating special cases that hacks the perceptual evaluator.

To prevent such hacking issue, we adopt the ideas of adversarial training. Adding the GAN loss and the perceptual term together successfully avoids such hacking. As shown in the last line of Table 6, adding all objectives achieves a consistent and signficant improvement in all metrics, demonstrating a real advancement in perceptual quality. It can be interpreted that the artifacts brought by overfitting on the metric related objectives can be easily spotted by the discriminator. With the adversarial supervision from the discriminator, the model is forced to learn to increase the metrics by improving generation quality.

We also conduct further ablation experiments regarding the design space of bridge modeling and EP-VAE training objectives. Results are shown in Appendix I due to page limit.

## 5 CONCLUSION

Our work presents VoiceBridge, modeling *diverse LQ-to-HQ tasks* in GSR with *a single latent-to-latent generative framework* backed by a transformer architecture. By introducing three novel techniques, including scale-equivariant regularization, joint neural prior, and perceptual-aware generation process, we shape a structured latent space for LBM, uniformly reduce the distance between each LQ prior and HQ target, and align both LBM sampling and VAE decoding with human perception, allowing VoiceBridge to consistently outperform strong GSR baselines on 48 kHz benchmarks and demonstrate strong performance on unseen tasks, *e.g.*, refining recent zero-shot speech, podcast, and dialogue generation methods. Ablations confirm the holistic improvements made by our innovations.

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

C ONTENTS

# A   RELATED WORK

## A.1   BRIDGE-BASED SPEECH ENHANCEMENT

Schrödinger Bridge (SB) models (Schrödinger, 1932; Chen et al., 2021; Wang et al., 2021; Bunne et al., 2023) explore learning an optimal stochastic trajectory between two boundary distributions with iterative fitting, allowing for efficient and high-quality generation. Built upon SB models, tractable bridge models have been recently developed (Bunne et al., 2023), which achieves *data-to-data* generation process effectively exploiting the instructive information contained in the observed prior distribution. These bridge models have shown advantages over diffusion models in tasks where indicative prior information has already been provided, such as image-to-image translation (Liu et al., 2023b), text-to-speech synthesis (Chen et al., 2023), image-to-video generation (Wang et al., 2024c), and speech restoration (SE) (Jukić et al., 2024).

In the field of SE, to exploit the advantages of the *data-to-data* generative process on SE tasks, recent works have designed bridge-based speech denoising, super-resolution, and dereveration models. In task-specific benchmark datasets, these works have proposed different innovations, such as designing the noise schedule (Jukić et al., 2024), examining the model parameterization (Chen et al., 2023), incorporating an adversarial training objective (Han et al., 2025), and introducing a two-stage generation framework (Wang et al., 2024a). Although these attempts have improved the generation quality and the inference speed of bridge-based SE systems, their applicability may still be restricted because of the narrow-band generation target, the specific degradation method, or the model training on a task-specific benchmark dataset.

In VoiceBridge, we propose a bridge-based general speech restoration system (GSR), restoring different low-quality (LQ) signals to the high-quality (HQ) target with a latent bridge model (LBM). By compressing the speech waveform into continuous latent representations, we model *diverse LQ-to-HQ restoration tasks* with *a unified latent-to-latent generative framework*, and propose three techniques to holistically strengthen the system, achieving GSR with high perceptual quality.

## A.2   GENERAL SPEECH RESTORATION

GSR, shown by (Liu et al., 2022), focuses on the task of retrieving clean speech out of lossy recordings, such as noise addition, bandwidth limitation, dereverberation, or other acoustic degradations. Unlike the traditional SE task which restores speech from only noise addition or another single kind of degradation, GSR pays extra focus on generating audio with high perceptual quality from acoustic scenarios with mixed degradations (Liu et al., 2022; Babaev et al., 2024).

Prior works explored different methods to solve the GSR task. As the initial proposer of GSR, Voice-Fixer (Liu et al., 2022) introduced a mapping-based method with two stages: firstly synthesizes the mel-spectrogram of clean audio from the distorted waveform, and then synthesizes the clean audio waveform through a vocoder. UniverSE (Serrà et al., 2022) and UniverSE++ (Scheibler et al., 2024) formulate the restoration as a diffusion generative process, where the degraded signal and the mel-spectrogram are taken as the condition information of diffusion models. Hi-ResLDM (Dhyani et al., 2025) is a two-stage model, including a mapping-based recovery stage and a diffusion-based restoration stage. FINALLY (Babaev et al., 2024) is an adversarial network, with additional alignment to self-supervised speech features and perceptual quality loss terms. MaskSR (Li et al., 2024) and AnyEnhance (Zhang et al., 2025b) are masked generative models, which generate the speech signal from discrete tokens. In VoiceBridge, we propose the first LBM-based GSR system, exploiting the two advantages of bridge models for GSR: full exploitation of the indicative LQ prior information, and iterative refinement nature along the entire sampling trajectory.

## B GSR TASK FORMULATION AND DATA CONSTRUCTION

In the formulation of GSR, the LQ speech sample is a degradation of the HQ target. The degradation operator $\mathcal{T}$ is a mixture of random degrading functions, i.e. $\mathcal{T} = \mathcal{T}_1 \circ \mathcal{T}_2 \circ \cdots \circ \mathcal{T}_N$, where each $\mathcal{T}_n$ is a specific degradation method including noise addition, bandwidth limitation, reverberation, clipping, etc. At the training stage of VoiceBridge, we follow previous works and construct the HQ and LQ speech data pairs with

$$\mathcal{T} = \mathcal{T}_{\text{bw}} \circ \mathcal{T}_{\text{clip}} \circ \mathcal{T}_{\text{rev}} \circ \mathcal{T}_{\text{noise}} \circ \mathcal{T}_{\text{rev}} \circ \mathcal{T}_{\text{eq}}, \tag{8}$$

where each degradation operator is applied with a certain probability, otherwise leaving the speech unchanged. These degradation operators are configured as follows.

- $\mathcal{T}_{\text{bw}}$: down-sampling the HQ speech sample to one of $\{2, 4, 8, 12, 16, 24, 32\}$ kHz (by uniformly random choice), with 0.5 probability. The down-sampling filter is chosen from the Bessel Filter, Chebyshev Filter, and Butterworth Filter with uniform randomness.
- $\mathcal{T}_{\text{clip}}$: clipping the HQ amplitude to the range $[0.06, 0.9]$ times of its original maximum absolute value, with 0.25 probability.
- $\mathcal{T}_{\text{rev}}$: applying room reverberation, with 0.5 probability. The reverberation is applied by convolving the speech signal with a room impulse response from a mixture of real-world and simulated RIR datasets.
- $\mathcal{T}_{\text{noise}}$: Adding noise with a random SNR in $[-5, 20]$ dB with 0.9 probability. Noise is randomly selected from the employed noise datasets
- $\mathcal{T}_{\text{eq}}$: Applying 1–3 random bell filters (frequency $\in [10, 12000]$ Hz, gain $\in [-5, 5]$ dB, $Q \in [0.5, 2]$, all sampled from a uniform distribution across the range) to each time window independently, with 0.5 probability.

Note that $\mathcal{T}_{\text{rev}}$ appears twice in Equation 8. This design follows (Zhang et al., 2025b), where the reverberation is applied twice, both inside and outside of noise addition, to ensure that additional noise involves audio pieces in both reverberant or non-reverberant environments. The parameters are chosen such that the degradation is noticeable by listeners.

## C SCHRÖDINGER BRIDGE TRAINING AND INFERENCE

The SB problem seeks the stochastic trajectory connecting two distributions $p_0$ and $p_1$ that minimizes the KL divergence to a reference diffusion process $p_{\text{ref}}$ (Schrödinger, 1932; Chen et al., 2021):

$$\min_{p \in \mathcal{P}[0,1]} \mathcal{D}_{\text{KL}}(p \| p_{\text{ref}}), \text{ subject to } p_0 = p_{\text{data}}, p_1 = p_{\text{prior}}. \tag{9}$$

where $\mathcal{P}[0, 1]$ is the space of path measures on the interval [0,1]. Here, we define $p_{\text{ref}}$ as the marginal distribution accompanying the forward SDE (Song et al., 2020):

$$d\mathbf{z}_t = f(\mathbf{z}_t, t)dt + g(t)d\mathbf{w}_t, \tag{10}$$

where $w_t$ is a standard Wiener process. Under this framework, the optimal SB dynamics are given by a pair of forward-backward SDEs with data-driven drift corrections:

$$d\mathbf{z}_t = \left[ f(\mathbf{z}_t, t) + g^2(t)\nabla \log \Psi_t(\mathbf{z}_t) \right] dt + g(t)d\mathbf{w}_t, \mathbf{z}_0 \sim p_0 \tag{11}$$

$$d\mathbf{z}_t = \left[ f(\mathbf{z}_t, t) - g^2(t)\nabla \log \bar{\Psi}_t(\mathbf{z}_t) \right] dt + g(t)d\bar{\mathbf{w}}_t, \mathbf{z}_1 \sim p_1 \tag{12}$$

where $\Psi_t$ and $\bar{\Psi}_t$ are solutions to the corresponding Schrödinger system PDEs, and their product defines the marginal density $p_t = \Psi_t \bar{\Psi}_t$ of the SB process. The drift term is typically chosen linearly as $f(\mathbf{z}_t, t) = f(t)\mathbf{z}_t$ with a predefined schedule $f(t)$. Although generally intractable, the SB problem has an analytical solution assuming that $p_0$ and $p_1$ are Gaussians centered on $\mathbf{z}_0$ and $\mathbf{z}_1$ (Chen et al., 2023; Bunne et al., 2023). Under this setting, the interpolated distribution $p_t$ at time $t$ is itself Gaussian:

$$p_t = \mathcal{N}\left( \frac{\alpha_t \bar{\sigma}_t^2}{\sigma_1^2} \mathbf{z}_0 + \frac{\bar{\alpha}_t \sigma_t^2}{\sigma_1^2} \mathbf{z}_1, \frac{\alpha_t^2 \bar{\sigma}_t^2 \sigma_t^2}{\sigma_1^2} \mathbf{I} \right) \tag{13}$$

where

$$\alpha_t = e^{\int_0^t f(\tau)d\tau}, \quad \bar{\alpha}_t = e^{-\int_t^1 f(\tau)d\tau}, \quad \sigma_t^2 = \int_0^t \frac{g^2(\tau)}{\alpha_\tau^2}d\tau, \quad \bar{\sigma}_t^2 = \int_t^1 \frac{g^2(\tau)}{\alpha_\tau^2}d\tau. \quad (14)$$

are noise schedules analytic from the reference SDE's $f, g$ functions (Chen et al., 2023).

Given the condition $\mathbf{z}_1$, the model is trained to predict unknown terms related to $(\mathbf{z}_t, t)$ in the reverse SB process (Equation 12) to generate $\mathbf{z}_0$ from $t = 1$ to $t = 0$. The model can be parameterized to predict either $\nabla \log \bar{\Psi}_t(\mathbf{z}_t)$ or other equivalent forms, similar to diffusion models. The most common and effective choice is to predict $\mathbf{z}_0$ directly and to minimize the mean square error (MSE) loss:

$$\mathcal{L}(\varphi) = \mathbb{E}_{(\mathbf{z}_0, \mathbf{z}_1)} \|\hat{\mathbf{z}}_{0,\varphi}(\mathbf{z}_t, t, \mathbf{z}_1) - \mathbf{z}_0\|_2^2, \quad (15)$$

where $\mathbf{z}_t$ is the noisy interpolation of $\mathbf{z}_0, \mathbf{z}_1$ on the bridge according to Equation 13.

After training, samples can be generated by simulating Equation 12 in reverse time, starting from condition $\mathbf{z}_1$. The sampling process can be principled accelerated by exponential integrators (Chen et al., 2023). Specifically, for the transition from $s$ to $t < s$, the first-order discretization takes the form

$$\mathbf{z}_t = \frac{\alpha_t \sigma_t^2}{\alpha_s \sigma_s^2}\mathbf{z}_s + \alpha_t \left(1 - \frac{\sigma_t^2}{\sigma_s^2}\right)\mathbf{z}_\theta(\mathbf{z}_s, s) + \alpha_t \sigma_t \sqrt{1 - \frac{\sigma_t^2}{\sigma_s^2}}\boldsymbol{\epsilon}, \quad \boldsymbol{\epsilon} \sim \mathcal{N}(0, \mathbf{I}) \quad (16)$$

$$\mathbf{z}_t = \frac{\alpha_t \sigma_t \bar{\sigma}_t}{\alpha_s \sigma_s \bar{\sigma}_s}\mathbf{z}_s + \frac{\alpha_t}{\sigma_1^2}\left[\left(\bar{\sigma}_t^2 - \frac{\bar{\sigma}_s \sigma_t \bar{\sigma}_t}{\sigma_s}\right)\mathbf{z}_\theta(\mathbf{z}_s, s) + \left(\sigma_t^2 - \frac{\sigma_s \sigma_t \bar{\sigma}_t}{\bar{\sigma}_s}\right)\frac{\mathbf{z}_1}{\alpha_1}\right] \quad (17)$$

for SDE and ODE, respectively.

# D  MODEL ARCHITECTURE AND TRAINING SETUP

## D.1  MODEL ARCHITECTURE

VoiceBridge consists of a VAE for prior latent encoding and target latent decoding, and a transformer backbone for solving the bridge model trajectory. For the autoencoder, we utilize the Oobleck VAE (Evans et al., 2024b) architecture. The encoder and decoder are symmetric and contain 156M parameters each. The VAE compresses 48 kHz audio into 64-channel latent representations at 23.4 Hz, yielding a temporal downsampling factor of 2048. The discriminator architecture follows (Evans et al., 2024a), where we employ a multi-scale STFT discriminator with 5 separate models modeling STFT spectrograms of different window lengths. For the SB model, we adopt the transformer backbone from Stable Audio 2 (Evans et al., 2024a;b), removing the text-conditioning cross-attention layers. We set the transformer backbone to have 24 transformer layers with a hidden dimension of 1152, resulting in 544 M parameters in total.

As Stable Audio 2 itself is a text-to-music generation model for a cross-modal audio generation task, we change its model architecture to GSR, which is a speech processing task. Specifically, we remove the text encoder and the cross-attention layers, reducing the model parameters and computation complexity. To include the low-quality audio as a condition, we concatenate the conditional LQ latent with the input latent on the channel dimension. Classifier-free guidance is also removed, as cross-modal generation is not involved.

## D.2  MODEL TRAINING DETAILS

The training pipeline of VoiceBridge consists of four stages: (1) pre-training a VAE with the Energy Preserving training objective $\mathcal{L}_{\text{ep-vae}}$ on clean speech; (2) fine-tuning the VAE encoder on distorted inputs with the joint neural prior loss $\mathcal{L}_{\text{np-enc}}$, while keeping the decoder frozen; (3) training a transformer-based bridge model to solve the SB in the latent space by $\mathcal{L}_{\text{bridge}}$, with both the encoder and decoder frozen; and (4) jointly fine-tuning the Bridge Transformer and the VAE decoder with $\mathcal{L}_{\text{pa-gen}}$, improving perceptual quality of generation process.

In the EP-VAE training section, the training objective has the form:

$$\mathcal{L}_{\text{ep-vae}} = \mathcal{L}_{\text{data}}^{\text{ep}}(\mathcal{D}_\theta(s \cdot \mathcal{E}_\theta(\mathbf{x})), s \cdot \mathbf{x}) + \mathcal{L}_{\text{latent}}(\mathcal{E}_\theta(\mathbf{x}), \mathbf{z}_{\text{ref}}), \quad (18)$$

where the $\mathcal{L}_{\text{data}}^{\text{ep}}$ and $\mathcal{L}_{\text{latent}}$ are the data-space and latent-space losses, respectively, consisting of the following loss terms:

$$\mathcal{L}_{\text{data}}^{\text{ep}} = \lambda_{\text{rec}}\mathcal{L}_{\text{rec}} + \lambda_{\text{adv}}\mathcal{L}_{\text{adv}} + \lambda_{\text{fm}}\mathcal{L}_{\text{fm}}, \tag{19}$$

$$\mathcal{L}_{\text{latent}} = \lambda_{\text{kl}}\mathcal{L}_{\text{kl}}, \tag{20}$$

in which $\mathcal{L}_{\text{rec}}, \mathcal{L}_{\text{adv}}, \mathcal{L}_{\text{fm}}$ and $\mathcal{L}_{\text{kl}}$ stand for the Multi-Resolution STFT reconstruction loss, adversarial loss, feature matching loss, and KL regularization loss, respectively. The loss weights $\lambda$ are used to balance the training objectives. In practice, we use the weights ensuring that each loss term has a similar order of magnitude, resulting in $\lambda_{\text{rec}} = 1, \lambda_{\text{adv}} = 0.1, \lambda_{\text{fm}} = 5, \lambda_{\text{kl}} = 1e - 4$. In this stage, the VAE is pre-trained on 8 A800 GPUs with a batch size of 16 for 800k steps, on the combined dataset for clean speech signals.

At the second stage, namely fine-tuning the encoder for the joint neural prior, we use the training objective

$$\mathcal{L}_{\text{np-enc}} = \mathcal{L}_{\text{data}}^{\text{ep}}(\mathcal{D}(s \cdot \mathcal{E}_\theta^{\text{np}}(\mathbf{x}_1)), s \cdot \hat{\mathbf{x}}_0) + \mathcal{L}_{\text{latent}}(\mathcal{E}_\theta^{\text{np}}(\mathbf{x}_1), \mathbf{z}_0), \tag{21}$$

for optimization. The $\mathcal{L}_{\text{data}}^{\text{ep}}$ term holds the same meaning as in Equation 19, while the $\mathcal{L}_{\text{latent}}$ now stands for the latent convergence objective that aligns different LQ prior latent with the ground-truth HQ target latent. In this stage, we hope to converge LQ prior latent representations in both scale and direction, thus designing both the MSE term and the cosine similarity term in the $\mathcal{L}_{\text{latent}}$ objective:

$$\mathcal{L}_{\text{latent}} = \lambda_{\text{mse}}\mathcal{L}_{\text{mse}} + \lambda_{\text{cos}}\mathcal{L}_{\text{cos}}, \tag{22}$$

where $\lambda_{\text{mse}} = \lambda_{\text{cos}} = 2.5$. We fine-tune the encoder from the initial state of the EP-VAE encoder, using paired data created with our construction pipeline introduced above. The fine-tuning process is conducted on 8 A800 GPUs with a batch size of 16 for 500k iterations.

At the third stage, we train the bridge transformer model to learn the bridge trajectory using Equation 15. This training stage is implemented on 32 A800 GPUs with a batch size of 256 for 1300k iterations.

At the last stage, we jointly fine-tune the Bridge Transformer and the VAE decoder to achieve higher perceptual quality. The training objective

$$\mathcal{L}_{\text{pa-gen}} = \mathcal{L}_{\text{bridge}}(\varphi) + \mathcal{L}_{\text{data}}(\mathcal{D}_\theta(\hat{\mathbf{z}}_{0,\varphi}(\mathbf{z}_t, t, \mathbf{z}_1^{\text{np}})), \mathbf{x}_0) + \mathcal{L}_{\text{hf}}(\mathcal{D}_\theta(\hat{\mathbf{z}}_{0,\varphi}(\mathbf{z}_t, t, \mathbf{z}_1^{\text{np}})), \mathbf{x}_0),$$

includes the bridge loss shown in Equation 15, VAE data-space loss shown in Equation 19, and two extra loss terms to align our generation with human perceptual quality:

$$\mathcal{L}_{\text{hf}} = \lambda_{\text{pesq}}\mathcal{L}_{\text{pesq}} + \lambda_{\text{utmos}}\mathcal{L}_{\text{utmos}}, \tag{23}$$

where $\mathcal{L}_{\text{pesq}}$ and $\mathcal{L}_{\text{utmos}}$ are two loss functions based on the PESQ and UTMOS scores, with $\lambda_{\text{pesq}} = 1$ and $\lambda_{\text{utmos}} = 10$. The joint finetuning is achieved on 8 A800 GPUs with a batch size of 16 for 200k iterations.

# E  DATASET DETAILS

We construct our HQ speech dataset by combining publicly available datasets: VCTK (Veaux & Yamagishi, 2017) is a multi-speaker English dataset with 109 speakers of diverse accents each reading about 400 sentences from newspaper excerpts and phonetic-rich passages, at a sampling rate of 48 kHz. HiFi-TTS (Bakhturina et al., 2021) contains speech from 10 English speakers reading LibriVox audiobooks and Project Gutenberg texts, specially filtered by a threshold of bandwidth and signal-to-noise ratio. HQ-TTS (Liu et al., 2022) is a collection of publicly available 44.1kHz or 48kHz clean speech data on the OpenSLR website, including multilingual speech from a wide range of speakers. AiShell-1 (Bu et al., 2017) and AiShell-4 (Fu et al., 2021) are two multi-speaker Mandarin datasets collected from noiseless indoor environments through different recording devices, providing non-English speech context for VoiceBridge to learn multilingual speech patterns. Bible-TTS (Meyer et al., 2022) is also a multilingual dataset, composed of speech for six languages spoken in Sub-Saharan Africa. Expresso (Nguyen et al., 2023) is a high-quality expressive speech dataset, which includes both expressively rendered read speech and improvised dialogues, offering speech in various tones and styles. EARS (Richter et al., 2024) contains anechoic speech recordings

Table 7: Bandwidth extension results on VCTK-BWE (Andreev et al., 2023). VoiceBridge beats other GSR models and audio super-resolution models on both subjective and objective metrics.

| Dataset | Model | WVMOS(↑) | NISQA(↑) | PESQ(↑) | STOI(↑) | LSD(↓) |
|---------|-------|----------|----------|---------|---------|--------|
| BW=1K | TFiLM | 1.650* | – | – | 0.810* | – |
| | HiFi++ | 3.710* | – | – | **0.860*** | – |
| | AudioSR | – | – | – | – | – |
| | NuWave2 | 1.895 | 2.958 | 1.671 | 0.751 | 1.762 |
| | VoiceFixer | 3.822 | 3.691 | 1.567 | 0.818 | 1.510 |
| | ResembleEnhance | 3.033 | 3.348 | 2.021 | 0.745 | 2.223 |
| | **VoiceBridge** | **4.154** | **4.58** | **2.11** | 0.824 | **1.433** |
| BW=2K | TFiLM | 2.270* | – | – | 0.910* | – |
| | HiFi++ | 3.950* | – | – | **0.940*** | – |
| | AudioSR | 3.435 | 4.176 | 1.752 | 0.907 | 1.539 |
| | NuWave2 | 3.208 | 3.755 | 2.098 | 0.885 | 1.580 |
| | VoiceFixer | 4.031 | 4.033 | 1.997 | 0.868 | 1.499 |
| | ResembleEnhance | 4.141 | 4.642 | 2.038 | 0.873 | 1.913 |
| | **VoiceBridge** | **4.306** | **4.656** | **2.696** | 0.897 | **1.392** |
| BW=4K | TFiLM | 3.490* | – | – | **1.000*** | – |
| | HiFi++ | 4.160* | – | – | **1.000*** | – |
| | AudioSR | 4.138 | 4.373 | 2.678 | 0.983 | 1.479 |
| | NuWave2 | 4.169 | 3.870 | 3.114 | 0.992 | 1.348 |
| | VoiceFixer | 4.144 | 4.290 | 2.471 | 0.909 | 1.507 |
| | ResembleEnhance | **4.439** | 4.684 | 2.685 | 0.941 | 1.845 |
| | **VoiceBridge** | 4.404 | **4.703** | **3.318** | 0.940 | **1.323** |

from over 100 English speakers with high demographic diversity, spanning the full range of human speech. Some of these datasets include speech samples recorded in different acoustic backgrounds, from which we select the clean audio in noiseless environments. For the datasets used for later evaluation, *e.g.*, VCTK, the corresponding testset is removed from the training data. The combination of training data, after the filtering, totals in around 1138 hours. All recordings are resampled to a sampling rate of 48 kHz.

For the noise addition and reverberation application processes of data construction, extra noise data and room impulse response (RIR) signals are required. For noise samples, we utilize the following datasets. DEMAND (Valentini-Botinhao, 2017) includes multi-channel recordings of acoustic noise in diverse environments. CHiME-2/3 (Vincent et al., 2013; Barker et al., 2015) have noise recordings taken from the two ChiME ASR challenges. TUT Urban Acoustic Scenes 2018 (Mesaros et al., 2018) is a dataset of multi-device recordings, originally used for urban acoustic scene classification. DESED (Turpault et al., 2019) is an audio dataset designed for recognizing sound event classes in domestic environments. These datasets have covered common noise in different acoustic scenarios, supporting our denoising-related tasks. Similar to the speech datasets, the testing portion is also removed for DEMAND and CHiME3, as they are later used for evaluation. Room Impulse Responses are drawn from the following datasets. SLR26/28 (Ko et al., 2017) contains simulated RIRs in different room conditions. GTU-RIR (Pekmezci & Genc, 2024) is a real RIR dataset, including the recordings of the impulse signal with different devices in different rooms. We also include other simulated RIR signals created by ourselves, following previous works.

Table 8: Dereverberation Results on WSJ-Reverb (Richter et al., 2023)

| | WSJ0–Reverb Welker et al. (2022) | | | |
|-------|--------|--------|--------|--------|
| Model | WVMOS (↑) | DNSMOS (↑) | NISQA (↑) | SRMR (↑) |
| SGMSE+ | 3.285 | 2.815 | 3.210 | 8.768 |
| StoRM | 3.583 | 2.969 | 3.823 | 9.615 |
| RE | 3.474 | 3.263 | 4.610 | 8.556 |
| VF | 3.093 | 3.056 | 3.849 | **10.278** |
| UniverSE++ | 2.572 | 2.313 | 3.131 | 8.472 |
| **VB(Ours)** | **4.403** | 3.286 | **4.617** | 9.929 |

Table 9: Dereverberation results on VCTK (Veaux & Yamagishi, 2017) convolved with simulated RIR signals with different RT60 time.

| Dataset | Method | PESQ(↑) | WVMOS(↑) | UTMOS(↑) | DNSMOS(↑) | NISQA(↑) | SRMR(↑) |
|---|---|---|---|---|---|---|---|
| RT60=0.3s | Voicefixer | 2.321 | 4.124 | 3.533 | _3.138_ | 4.369 | 9.044 |
| | Resemble-Enhance | 2.333 | **4.376** | _3.628_ | 3.101 | _4.381_ | 9.208 |
| | UniverSE++ | **2.599** | 4.102 | 3.296 | 3.010 | 4.024 | _9.744_ |
| | **VoiceBridge** | _2.393_ | _4.286_ | **3.864** | **3.185** | **4.488** | **9.767** |
| RT60=0.6s | Voicefixer | _2.060_ | 4.068 | 3.394 | **3.111** | 4.294 | 9.398 |
| | Resemble-Enhance | 1.653 | **4.244** | _3.419_ | 3.057 | _4.317_ | 9.208 |
| | UniverSE++ | 1.652 | 3.553 | 2.639 | 2.830 | 3.695 | **9.794** |
| | **VoiceBridge** | **2.127** | _4.242_ | **3.821** | _3.081_ | **4.448** | _9.762_ |
| RT60=0.9s | Voicefixer | _1.915_ | 4.024 | _3.342_ | **3.108** | 4.292 | _9.641_ |
| | Resemble-Enhance | 1.394 | _4.139_ | 3.232 | _3.067_ | **4.353** | 9.207 |
| | UniverSE++ | 1.407 | 3.251 | 2.329 | 2.755 | 3.602 | 9.604 |
| | **VoiceBridge** | **1.970** | **4.195** | **3.757** | 3.058 | _4.358_ | **9.919** |
| RT60=1.2s | Voicefixer | _1.837_ | 3.995 | _3.292_ | **3.105** | 4.261 | _9.658_ |
| | Resemble-Enhance | 1.323 | _4.100_ | 3.175 | 3.051 | _4.271_ | 9.334 |
| | UniverSE++ | 1.351 | 3.170 | 2.193 | 2.706 | 3.540 | 8.999 |
| | **VoiceBridge** | **1.873** | **4.177** | **3.732** | _3.058_ | **4.338** | **9.968** |

# F  EVALUATION DETAILS

In this section, we provide a detailed introduction to the benchmark datasets, baseline methods, and evaluation metrics that we used for evaluating VoiceBridge's performance. We also list further evaluation settings and results on broken down degradation types in detail, and make comparison to other models.

## F.1  EVALUATION DATASETS

To comprehensively evaluate VoiceBridge's GSR performance at scale, we utilized three test datasets as follows. VoiceFixer-GSR (Liu et al., 2022) is a simulated testset, which produces LQ speech samples with a random degradation process combining noise addition, low-pass filtering, clipping, and reverberation. We utilize this benchmark to evaluate VoiceBridge's multi-degradation restoration ability and to enable direct comparison with VoiceFixer.

DNS-with-Reverb (Reddy et al., 2021a) augments DNS Challenge samples with reverberation and noise, generating a set of noisy audio in strong reverberant conditions. This testset evaluates VoiceBridge's performance under challenging degradation scenarios. Meanwhile, *all training data samples from the DNS Challenge are excluded from the training dataset of VoiceBridge, causing an extra training-inference gap, and testing VoiceBridge's robustness in unseen conditions.*

DNS-Real-Data (Reddy et al., 2021a) is an in-the-wild dataset, composed of real recordings from the DNS Challenge. It contains lossy audio samples recorded in a real acoustic environment, demonstrating the real-world restoration abilities of VoiceBridge.

As GSR is defined as general restoration, including different restoration processes, the restoration process from a single degradation naturally becomes its subtask. To evaluate VoiceBridge's zero-shot ability on these restoration subtasks, we choose the traditional Speech Enhancement (SE) task as a typical example. We utilize VoiceBank-Demand (Valentini-Botinhao, 2017) and WSJ0-CHiME3 (Barker et al., 2015). The VoiceBank-Demand dataset mixes utterances from the Voicebank speech corpus with real-world noise from the DEMAND dataset, serving as a benchmark testset for the SE. WSJ0-CHiME3, similarly, mixes clean speech signals from the WSJ0 dataset with noise from CHiME3, but with a smaller signal-to-noise ratio, making the denoising task harder. The two benchmark datasets are widely used in prior SE works (Lu et al., 2021; 2022; Richter et al., 2023; Lemercier et al., 2023b; Jukić et al., 2024).

To further verify the zero-shot performance of bridge models, we report VoiceBridge's performance on *Out-of-Domain* (OOD) tasks. We include test results for codec artifact removal (CAR), and text-to-speech (TTS) refinement, which are especially significant for contemporary TTS applications.

Table 10: Denoising results on VCTK (Veaux & Yamagishi, 2017) with additive noise from DEMAND (Thiemann et al., 2013) at different SNR levels.

| Dataset | Method | PESQ(↑) | WVMOS(↑) | UTMOS(↑) | DNSMOS(↑) | NISQA(↑) |
|---|---|---|---|---|---|---|
| SNR=-15dB | SBSE | 1.119 | -0.949 | 1.480 | 1.513 | 1.869 |
| | Voicefixer | 1.225 | 2.423 | 2.278 | 2.313 | 2.848 |
| | Resemble-Enhance | 1.168 | 3.394 | 2.440 | 2.640 | 3.130 |
| | UniverSE++ | 1.211 | 2.772 | 2.271 | 2.374 | 2.950 |
| | **VoiceBrigde** | **1.352** | **3.479** | **2.755** | **2.698** | **3.270** |
| SNR=-10dB | SBSE | 1.224 | 0.347 | 1.829 | 1.822 | 2.389 |
| | Voicefixer | **1.588** | 3.263 | 2.845 | 2.791 | 3.576 |
| | Resemble-Enhance | 1.345 | 3.783 | 2.929 | 2.876 | 3.693 |
| | UniverSE++ | 1.527 | 3.704 | 3.075 | 2.933 | **3.971** |
| | **VoiceBrigde** | 1.453 | **3.908** | **3.339** | **2.944** | 3.725 |
| SNR=-5dB | SBSE | 1.410 | 1.698 | 2.301 | 2.275 | 2.933 |
| | Voicefixer | **1.871** | 3.901 | 3.346 | **3.106** | 4.133 |
| | Resemble-Enhance | 1.586 | 4.001 | 3.266 | 3.009 | 4.075 |
| | UniverSE++ | 1.790 | 3.820 | 3.320 | 2.913 | **4.177** |
| | **VoiceBrigde** | 1.730 | **4.093** | **3.641** | 3.022 | 3.969 |
| SNR=0dB | SBSE | 1.663 | 2.938 | 2.826 | 2.665 | 3.468 |
| | Voicefixer | 2.143 | 4.074 | 3.548 | **3.170** | **4.350** |
| | Resemble-Enhance | 1.891 | 4.135 | 3.491 | 3.070 | 4.322 |
| | UniverSE++ | **2.240** | 4.100 | 3.674 | 2.974 | 4.291 |
| | **VoiceBrigde** | 2.200 | **4.200** | **3.765** | 3.098 | 4.201 |

For the CAR test, we utilize the VCTK (Veaux & Yamagishi, 2017) testset, using Encodec (Défossez et al., 2022) to compress the 48kHz audio to a codec at a 3kbps frame rate, and then reconstruct it.

For the TTS task, we choose the Seed-TTS (Anastassiou et al., 2024) benchmark, which is widely used in various TTS works (Wang et al., 2024b; Ju et al., 2025; Boson AI, 2025; Peng et al., 2025), and generate speech and podcast using MaskGCT (Wang et al., 2024b) and MoonCast (Ju et al., 2025) respectively. Note that the MaskGCT is used with a prompt speech for the speaker condition, and MoonCast is called without a prompt audio.

## F.2 BASELINE METHODS

As a GSR model, we compare VoiceBridge against other strong GSR baseline methods for both GSR tasks and OOD subtasks. VoiceFixer (Liu et al., 2022) is a mapping-based model with two stages: synthesizes the mel-spectrogram of the clean speech from the distorted one at the first stage, and then generates the clean audio waveform through a vocoder. As the first proposer of the GSR task, it stands as a classic baseline to compare with. Resemble-Enhance (Niu, 2024) is an open-sourced GSR tool based on the flow-matching generative process, and a popular baseline model used in several GSR works. UniverSE++ (Scheibler et al., 2024) is an improved version of UniverSE (Serrà et al., 2022), which formulates the restoration as a conditional diffusion generation process, where the degraded observation is taken as condition information. We use the pre-trained checkpoints of these models for inference on the benchmark test sets mentioned above. AnyEnhance (Zhang et al., 2025b) is a masked generative model that demonstrates strong performance on GSR. We use their reported values in their paper due to a lack of publicly available implementation.

To further demonstrate the performance of VoiceBridge on SE, we include not only the GSR models listed above, but also strong task-specific baseline methods for comparison. For SE, we include the following baseline models: SGMSE+ (Richter et al., 2023; Welker et al., 2022) is a simple diffusion model used in both denoising and dereverberation scenarios, but needs to be trained separately. StoRM (Lemercier et al., 2023b) makes further improvement based on SGMSE+, adding an extra predictive stage before diffusion generation, reducing the gap between diffusion target and prior conditions. SBSE (Jukić et al., 2024) is an SB model trained for the SE task, demonstrating data-space SB performance. Noticeably, we use the pre-trained checkpoint of these models on VoiceBank-Demand, WSJ0-CHiME3, or WSJ0-Reverb training set, and report the inference result

on the corresponding test sets, whereas VoiceBridge is used in a cross-validation way where the training set is composed of other audios.

## F.3 METRICS

To make a comprehensive and fair comparison, we use both the intrusive and the non-intrusive metrics to evaluate our GSR performance. As to intrusive metric, we use PESQ (Rix et al., 2001) which is a widely adopted ITU-T standard. It was developed to model the subjective test and works at a 16 kHz sampling rate. Nowadays, non-intrusive metrics measure perceptual quality without reference signals, and are believed to be more relevant to generative SE (Manjunath, 2009; Manocha et al., 2022; Babaev et al., 2024). We follow previous GSR works and adopt several non-intrusive metrics for comprehensive evaluation. DNSMOS (Reddy et al., 2021b) is based on a multi-stage CNN trained on human MOS scores. It takes the log Mel spectrogram as the input feature and predicts the estimated signal quality, background noise quality, and overall quality (SIG, BAK, OVRL) scores. WV-MOS is (Ogun et al., 2023) is a non-intrusive metric based on self-supervised wav2vec2 (Baevski et al., 2020) audio features. UTMOS (Saeki et al., 2022) is an evaluation system based on ensemble learning of strong and weak learners, achieving high overlap with human opinion on both in-domain and out-of-domain data. All the above metrics operate at 16 kHz, where the inference results are first down-sampled before metric calculation. To evaluate the perceptual quality of the high-frequency components in the generated full-band audio, we include NISQA (Mittag et al., 2021), a quality prediction model that takes 48 kHz audio as input.

## F.4 DEGRADATION BREAKDOWN

To evaluate the zero-shot capability of VoiceBridge and baseline models across various types and degrees of degradations, we further break down several degradation type into different levels by severity, either following already proposed benchmarks or curating new testsets, and report the performance. For the Speech Bandwidth Extension (BWE) task, we follow previous works and utilize the VCTK-BWE (Andreev et al., 2023) benchmark, which is composed of VCTK speech samples with bandwidth limited to 1,2,4 kHz Nyquist frequencies, where models are required to generate full-band (*i.e.*, 48kHz) target from the severely down-sampled signals. Such BWE task settings are aligned with the setting of bandwidth extension in other speech restoration works (Andreev et al., 2023; Lemercier et al., 2023a; Ku et al., 2025), where the lowpass filtering causes severe information loss in the speech signals, challenging the models to restore high-quality speech using limited prior knowledge. For the Dereverberation (Derev) task, we report experiment results on the WSJ-Reverb (Richter et al., 2023) benchmark, which constructs reverberant signals from speech in the WSJ0 corpora and the simulated room-impulse-responses. This benchmark is utilized in prior speech dereverberation works (Jukić et al., 2024). Meanwhile, to test model performance under different severity of reverberations, we create another test set by adding simulated RIR signals to speech from the VCTK dataset, with the RT60 time (time needed for sound energy to decay by 60 dB) ranging in 0.3, 0.6, 0.9, and 1.2 seconds. For the SE task, we still utilize the VoiceBank corpus and the DEMAND noise dataset, but this time adding noise at a fixed signal-to-noise ratio (SNR). We curated 6 testsets, with SNR at -15,-10,-5,0dB respectively, covering a wide range of additive noise from slight disturbance to severe degradation.

To make comprehensive evaluation, we include task-specific baseline models and evaluation metrics. For the BWE task, we include audio super-resolution models and other speech BWE model, including AudioSR (Liu et al., 2024), NuWave2 (Han & Lee, 2022), TFiLM (Birnbaum et al., 2019), and HiFi++ (Andreev et al., 2023). For dereverberation on WSJ-Reverb, we include SGMSE+ (Richter et al., 2023; Welker et al., 2022) and StoRM (Lemercier et al., 2023b). For SE, we include SBSE (Jukić et al., 2024). Additional metrics involves the following: STOI (Taal et al., 2010) is an objective metric that measures the intelligibility of degraded speech. LSD (Liu et al., 2022) directly calculates the log-spectrogram-distance between the enhanced and ground truth signals, which is a commonly used metric in BWE scenarios (Liu et al., 2024; Li et al., 2025a). SRMR (Falk et al., 2010), a non-intrusive metric specially designed for measuring signal to reverberation energy ratio, is utilized for measuring dereverberation performance in our experiments.

The results are are listed in Table 7,Table 8, Table 9 and Table 10. For the bandwidth extension results shown in Table 7, VoiceBridge consistently outperforms baseline models on the NISQA,

PESQ, and LSD metrics, outperforming the specifically trained bandwidth extension models and other generalist models. For the dereverberation results shown in Table 8, VoiceBridge achieves the highest WVMOS, DNSMOS, and NISQA score, and second-best SRMR score, showing strong ability despite the cross-dataset setting. In the detailed experiments of reverberation and additional noise at different severities, shown in Table 9 and Table 10, VoiceBridge exhibits strong performance in all settings, including those with degradation severity well beyond the training data. These results demonstrate VoiceBridge's strong zero-shot performance across various Speech Restoration scenario.

Table 11: Comparison with Metis (Wang et al., 2025) on the two DNS testsets. Note that Metis used 300 times more data than hours for pretraining, and meanwhile generating only 24kHz audio, thus gaining better results in DNSMOS metrics (which operate at 16kHz). NISQA provides more comprehensive evaluation by taking full-band audio as input.

| Dataset | Method | SIG($\uparrow$) | BAK($\uparrow$) | OVRL($\uparrow$) | NISQA($\uparrow$) |
|---|---|---|---|---|---|
| DNS with Reverb | Metis | 3.68 | 4.14 | 3.44 | 4.56 |
| | **VoiceBridge** | 3.55 | 4.05 | 3.27 | 4.56 |
| DNS Real | Metis | 3.59 | 4.01 | 3.27 | 3.95 |
| | **VoiceBridge** | 3.45 | 3.99 | 3.17 | 4.50 |

### F.5 COMPARISON WITH PRETRAINED MODELS

We further compare VoiceBridge to contemporary generative pretrained models. Metis (Wang et al., 2025), is a foundational speech model pretrained on 300 K hours of speech data using a masked-generative strategy. It can then be fine-tuned to multiple generative speech tasks, including Speech Restoration. We report the comparison of Metis and VoiceBridge on the two DNS testsets (Reddy et al., 2021a) (DNS-with-Reverb and DNS-Real) in Table 11. VoiceBridge exhibits slightly lower performance than Metis on DNSMOS metrics but achieves comparable results on NISQA under simulated test conditions and substantially surpasses Metis in real-world NISQA evaluations. This discrepancy can be attributed in part to the characteristics of the evaluation metrics: DNSMOS operates only at 16 kHz and therefore fails to capture fine-grained high-frequency details. Metis, which generates speech at 24 kHz, benefits from modeling a narrower frequency range and thus has an inherent advantage under DNSMOS evaluation. By contrast, NISQA provides a more comprehensive assessment, as it considers full-band audio. Furthermore, Metis was trained with approximately 300 times more pre-training data, much of which is closed-source (Wang et al., 2025). In comparison, VoiceBridge achieves competitive performance using only public datasets, highlighting its superior data efficiency.

### F.6 COMPARISON WITH OTHER CLOSED-SOURCE MODELS

Moreover, we compare VoiceBridge to other closed-source models such as FINALLY Babaev et al. (2024), HiFi-GAN-2 Su et al. (2021), Speech-Flow Liu et al. (2023a), and Ku et al. Ku et al. (2025). For comparison with FINALLY

Table 12: Comparison with FINALLY (Babaev et al., 2024) and HiFi-GAN-2 Su et al. (2021), with samples from FINALLY's demo page.

| FINALLY (Babaev et al., 2024) Demonstration Samples | | | |
|---|---|---|---|
| Model | DNSMOS ($\uparrow$) | UTMOS ($\uparrow$) | WV-MOS ($\uparrow$) |
| HiFi-GAN-2 | 3.283 | 3.947 | 3.470 |
| FINALLY | **3.313** | 3.847 | 3.989 |
| VoiceBridge | 3.063 | **4.111** | **4.034** |

and HIFI-GAN-2, we use the open samples on the demonstration page of FINALLY at https://mmacosha.github.io/finally-demo/. The results are shown in Table 12. We can see that, VoiceBridge beat FINALLY and HiFi-GAN-2 on the WV-MOS and UTMOS metric, while reaching on-par performance on the DNSMOS metric, demonstrating competitive performance against strong closed-source baselines. We need to note that, FINALLY is **pretrained on large internal** datasets, where as VoiceBridge is only trained on publicly available data, which highlights the advantage of our modeling. Meanwhile, the testing samples are from the FINALLY's

demo page, which might be cherry-picked results for the baseline model, yet completely random for VoiceBridge.

To compare against Speech-Flow and Ku et al.'s model, we evaluated VoiceBridge on the benchmark and metrics used in the corresponding papers. We reported the evaluation results on two denoising benchmarks VB-Demand (Valentini-Botinhao, 2017) and WSJ0-CHiME3 (Barker et al., 2015), and the bandwidth extension benchmark WSJ0-BWE (Ku et al., 2025). The results are shown in Table 13. VoiceBridge achievew comparable performance with the two baseline models on non-intrusive metrics, such

Table 13: Comparison with SpeechFlow (Liu et al., 2023a) and Ku et al. Ku et al. (2025), on corresponding benchmarks.

| Downstream Task Evaluation on Curated Testsets | | | |
|---|---|---|---|
| VB-Demand | | | |
| Model | PESQ (↑) | WVMOS (↑) | ESTOI (↑) | SISDR (↑) |
| SpeechFlow | 3.13 | - | 0.87 | - |
| Ku et al. | **3.27** | **4.41** | **0.88** | **19.1** |
| VoiceBridge | 2.82 | 4.39 | 0.79 | 4.8 |
| WSJ0-CHiME3 | | | |
| Model | PESQ (↑) | WVMOS (↑) | ESTOI (↑) | SISDR (↑) |
| Ku et al. | **2.85** | 4.26 | **0.92** | **16.2** |
| VoiceBridge | 1.72 | **4.36** | 0.82 | 3.4 |
| WSJ0-BWE | | | |
| Model | WVMOS (↑) | SQUIM-MOS (↑) | ESTOI (↑) | SISDR (↑) |
| Ku et al. | 3.93 | 4.41 | **0.92** | **8.5** |
| VoiceBridge | **4.28** | **4.48** | 0.85 | 3.5 |

as WVMOS and SQUIM-MOS, but showcase lower performance on other metrics. For this, we must state that: SpeechFlow and Ku et al.'s methods need specific finetuning for each task, while VoiceBridge is a general model without any task-specific adaptation. Meanwhile, both SpeechFlow and Ku et al.'s models work on 16kHz audio, while VoiceBridge generates 48kHz full-band audio, which brings significantly more challenges for the increased data dimension and high-frequency modeling. The used metrics only evaluate the lower 16kHz band, bringing inherent disadvantages to VoiceBridge. On waveform-level metrics like SISDR, latent-space methods like VoiceBridge typically underperforms data-level methods due to the latent reconstrucion error (Especially, VAE reconstruction is trained mainly on MR-STFT maginitude, not on the waveform dimension, and no phase preservation is guaranteed) and the cascading error from latent modeling. However, these metrics do not necessarily reflect the generated audio quality, as pointed out in (Manocha et al., 2022; Manjunath, 2009; Andreev et al., 2023; Babaev et al., 2024).

## G FURTHER STUDIES ON LATENT MODELING

### G.1 QUANTITATIVE RESULTS ON THE JOINT NEURAL PRIOR

To quantitatively analyze the effect of prior convergence beyond tSNE visualization, we calculate the 2-Wasserstein distance (Panaretos & Zemel, 2019) matrix over the latent distribution for different priors. We model the latent distribution for each degradation type as a Gaussian distribution with the sample mean and variance. The 2-Wasserstein (Fréchet) distance between two Gaussian-fitted distributions $\mathcal{N}(\mu_i, \Sigma_i)$ and $\mathcal{N}(\mu_j, \Sigma_j)$ is defined as

$$W_2^2\big(\mathcal{N}(\mu_i, \Sigma_i), \mathcal{N}(\mu_j, \Sigma_j)\big) = \|\mu_i - \mu_j\|_2^2 + \mathrm{Tr}\Big(\Sigma_i + \Sigma_j - 2\big(\Sigma_i^{1/2}\Sigma_j\Sigma_i^{1/2}\big)^{1/2}\Big). \quad (24)$$

The resulting matrix represents all pairwise distances between the latent distributions of different degradation types. For the diagonal entries, we use a within-label variance proxy by measuring the Wasserstein distance to a Dirac at the mean, which equals the square root of the total variance:

$$W_2\big(\mathcal{N}(\mu_c, \Sigma_c), \delta_{\mu_c}\big) = \sqrt{\mathrm{Tr}(\Sigma_c)}. \quad (25)$$

As shown in the Figure 5, The joint neural prior yields closer distance between priors of different distributions than the vanilla priors. Moreover, the Wasserstein distance between different degradation types converges toward the diagonal terms, indicating that all priors converge to a single distribution.

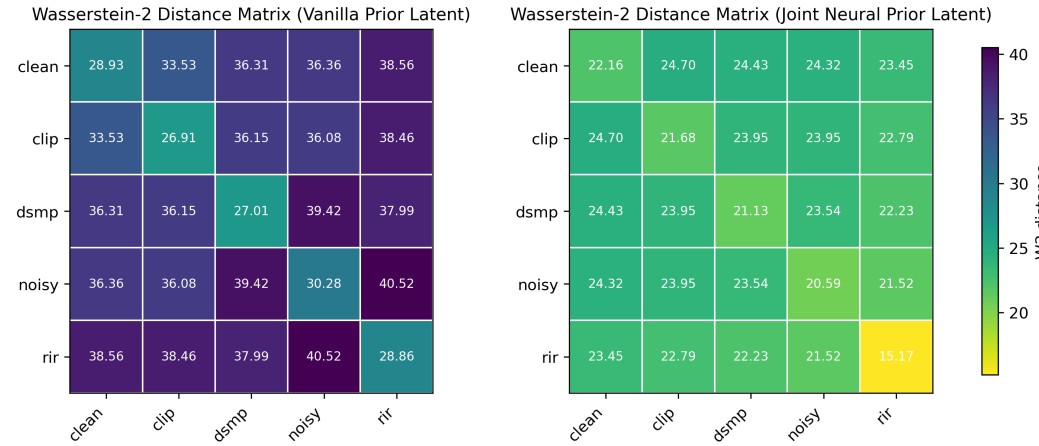

Figure 5: Wasserstein Distance Matrix of latents with different degradation types for the vanilla prior and the joint neural prior.

## G.2 EXCEEDING THE PERCEPTUAL QUALITY LIMITS OF LATENT MODELS

Without our proposed perceptual-aware fine-tuning process, the LBM is trained to capture the distribution of the latent of the HQ target, which may not be fully aligned with human perceptual quality. Moreover, without joint fine-tuning, the perceptual generation quality of VoiceBridge will be limited by VAE as well. As the perceptual-aware objectives have not been explicitly considered in the training of VAE, the perceptual quality of VoiceBridge will be inevitably restricted. By the perceptual-aware joint fine-tuning process, we can improve the perceptual generation quality of both LBMs and the VAE decoder, meanwhile reducing their cascading error in generation.

Table 14: Comparison of perceptual quality between VAE reconstruction, VoiceBridge restoration, VoiceBridge's reconstruction results on clean audio input, and Ground Truth clean audio.

|          | DNSMOS(↑) | WVMOS(↑) | NISQA(↑) |
| -------- | --------- | -------- | -------- |
| VAE-Rec. | 3.11      | 4.05     | 4.5      |
| VB-Gen.  | 3.20      | 4.39     | 4.51     |
| VB-Rec.  | **3.25**  | _4.43_   | **4.71** |
| GT       | _3.22_    | **4.5**  | _4.62_   |

To verify that VoiceBridge surpasses the upper-bound reconstruction performance of the original VAE, we compare the restoration results of noisy inputs with EP-VAE reconstructions of clean ones. Specifically, we report four groups of results: (1) the EP-VAE reconstruction of clean audio (VAE-Rec.); (2) the restoration results of corresponding degraded audio of VoiceBridge (VB-Gen.); (3) the restoration results of VoiceBridge when passed the clean audio as input (VB-Rec.), which can be seen as a version of reconstruction with enhanced perceptual quality by VoiceBridge; (4) the ground-truth clean target (GT). Clean and degraded data are taken from the VoiceBank-Demand (Valentini-Botinhao, 2017) dataset. As shown in Table 14, the perceptual-aware fine-training stage enables VoiceBridge's restoration to surpass the VAE reconstruction limits in perceptual quality. Moreover, when GT is passed as input for VoiceBridge instead of its noisy version, the perceptual quality further improves, achieving near-GT or even better performance on the reconstruction setting. This indicates that jointly training the LBM and the decoder together breaks through the VAE reconstruction upper-bound.

## G.3 VAE-ONLY RESTORATION

In the joint neural prior training stage, the VAE is trained to align the degraded prior to clean prior in multiple measures, including those measuring reconstruction distances, which means the reconstruction of degraded speech will move forward the direction of the restoration target. After the

Table 15: VAE-only restoration results on different degradations

| | Curated Validation Sets for Different Degradations | | | | |
|---|---|---|---|---|---|
| Model | PESQ (↑) | DNSMOS (↑) | WVMOS (↑) | UTMOS (↑) | NISQA (↑) |
| *Noise* | | | | | |
| VoiceBridge | **2.34** | **3.17** | **4.30** | **4.26** | **4.18** |
| VAE | 1.71 | 2.58 | 3.51 | 3.37 | 3.23 |
| *Reverb* | | | | | |
| VoiceBridge | **2.67** | **3.24** | **4.41** | **4.31** | **4.63** |
| VAE | 1.52 | 2.72 | 3.85 | 3.69 | 4.20 |
| *BWE* | | | | | |
| VoiceBridge | **3.32** | **3.23** | **4.41** | **4.31** | **4.64** |
| VAE | 2.63 | 2.83 | 3.75 | 3.79 | 4.41 |
| *CLIP* | | | | | |
| VoiceBridge | **3.57** | **3.23** | **4.43** | **4.31** | **4.69** |
| VAE | 2.94 | 2.82 | 3.96 | 3.90 | 4.44 |

perceptual-aware finetuning stage, the decoder is further adapted to generate speech with high perceptual quality. This indicate that the VAE itself have certain amount of restoration abilities. To further evaluate the ability of the VAE and the role of the latent bridge model, we test the performance of VAE-only restoration compared with VoiceBridge on different degradation types. The results are shown in Table 15. Basically, we can find that VAE-only enhancement works well on Bandwidth extension task and de-clipping tasks. These two tasks are considered easier, since they preserve the main structure of the speech signal. For the other harder tasks, generative modeling gives more effort. The bridge model successfully refines all degradations to a similar HQ leel beyond the VAE reconstruction.

Table 16: Evaluation results on a subset of VoiceFixer-GSR (Liu et al., 2022) with different sampling steps of VoiceBridge

| Steps | WV-MOS (↑) | UTMOS (↑) | DNSMOS (↑) | NISQA (↑) | PESQ (↑) |
|---|---|---|---|---|---|
| steps=1 | 4.011 | 3.965 | 2.905 | 3.846 | 2.378 |
| steps=2 | 4.105 | 4.062 | 3.005 | 3.924 | 2.522 |
| steps=3 | 4.182 | 4.364 | 3.158 | 4.156 | 2.589 |
| steps=4 | **4.291** | **4.367** | **3.205** | 4.160 | **2.692** |
| steps=5 | 4.269 | 4.354 | 3.193 | 4.217 | 2.545 |
| steps=10 | 4.201 | 4.358 | 3.200 | **4.296** | 2.445 |
| steps=20 | 4.189 | 4.329 | 3.179 | 4.188 | 2.365 |
| steps=50 | 4.145 | 4.330 | 3.161 | 4.269 | 2.270 |

## H   INFERENCE RESULTS WITH DIFFERENT SAMPLING STEPS

We test our GSR inference efficiency on a subset of VoiceFixer-GSR (Liu et al., 2022) with different sampling steps. As shown in Table 16, the best restoration results are achieved with around 4 sampling steps, similar to the observation in (Jukić et al., 2024). In the field of SE, a large number of sampling steps may not lead to better results, which has been observed in both diffusion and bridge-based models. In our analysis, one of the reasons is that, as the prior or condition information (namely the LQ observation) has already been very indicative of the HQ target, diffusion or bridge models are not required to generate the target with numerous iterative sampling steps. Given a large number of sampling steps, some of the sampling iterations may introduce error and result in sub-optimal generation quality.

Table 17: Inference efficiency, evaluated by Real-Time-Factor (RTF) and Number of Function Evaluation (NFE)

| Model | VF | RE | UPP | **VB** |
|---|---|---|---|---|
| RTF ($\downarrow$) | 0.015 | 0.833 | 0.075 | 0.068 |
| NFE | 1 | 64 | 8 | 4 |

# I  DETAILS ON ABLATION STUDIES

## I.1  EXPERIMENTAL SETUP

We conduct ablation studies on each innovation of VoiceBridge: EP-VAE, joint neural prior, and perceptual-aware post-training. For the first part of the ablation study, which mainly focuses on the variation of VAE networks, the VAE networks (with or without EP) are pre-trained for 500k steps on 8 A800 GPUs with a batch size of 16. The fine-tuning for joint neural prior costs 150k steps with the same settings. For fair comparison, the groups without joint neural prior are trained using the original objective for the same amount of steps. For the second part, which studies the different loss terms in the post-training stage, all variants are fine-tuned from the pre-trained bridge model in the main experiment for 50k steps on 8 A800 GPUs with a batch size of 16.

Table 18: Evaluation results on VoiceFixer-GSR (Liu et al., 2022) and DNS-Real (Reddy et al., 2021a) for bridge modeling in different spaces.

| | Voicefixer-GSR (Liu et al., 2022) | | | | | | |
|---|---|---|---|---|---|---|---|
| Model | PESQ ($\uparrow$) | SIG ($\uparrow$) | BAK ($\uparrow$) | OVRL ($\uparrow$) | UTMOS ($\uparrow$) | WV-MOS ($\uparrow$) | NISQA ($\uparrow$) |
| Waveform SB | 1.56 | 2.65 | 3.83 | 2.41 | 2.21 | 2.87 | 1.73 |
| STFT SB | 2.06 | 3.07 | 3.34 | 2.58 | 2.69 | 2.44 | 2.73 |
| Latent SB | **2.07** | **3.43** | **4.09** | **3.20** | **3.58** | **3.86** | **3.76** |
| | DNS-Real (Reddy et al., 2021a) | | | | | | |
| Model | PESQ ($\uparrow$) | SIG ($\uparrow$) | BAK ($\uparrow$) | OVRL ($\uparrow$) | UTMOS ($\uparrow$) | WV-MOS ($\uparrow$) | NISQA ($\uparrow$) |
| Waveform SB | - | 2.64 | 2.55 | 1.94 | 1.61 | 1.55 | 1.56 |
| STFT SB | - | 2.96 | 3.57 | 2.55 | 2.27 | 2.59 | 3.39 |
| Latent SB | - | **3.39** | **4.03** | **3.12** | **3.15** | **3.69** | **4.34** |

## I.2  ABLATION ON BRIDGE MODELING SPACE

To validate the necessity of latent modeling for the GSR task, we further conduct an ablation study on the modeling space for SB models. We train three bridge models on VAE latent, waveform, and complex STFT data space respectively. For the waveform-space bridge, we adopt the diffwave () architecture. For the STFT space, the NeMo () (adopted in () and ) architecture is used. All networks are scaled to around 540M parameters by increasing the number of layers and hidden channels. For the VAE latent, we use VoiceBridge's configuration without the perceptual-aware finetuning process, for equal comparison on the modeling spaces. All models are trained for 500k steps on the same data mixture as VoiceBridge. We evaluate the three models on the VoiceFixer-GSR and the DNS-Real benchmarks. The results are shown in Table 18 We can clearly observe that latent bridge models far outperform waveform and STFT domain bridge models, given the high information density of the compressed latent space. Meanwhile, the latent SB models take much less computation than the other two models, as the sequence modeled is much shorter.

## I.3  ABLATION ON JOINT NEURAL PRIOR TRAINING OBJECTIVE DESIGN

A core difference of our joint neural prior with (Fang et al., 2021) is that we align the LQ and HQ priors not only by the Euclidean measure in latent space (wich is basically equivalent with the KL alignment done in (Fang et al., 2021) if regardless of the variance). In contrast, we add objectives including cosine similarity in latent space and multiple supervision signals in the reconstructed data space, in order to reduce the distance between the latents in all geometrical measures. We conduct

Table 19: Evaluation results on VoiceFixer-GSR (Liu et al., 2022) and DNS-Real (Reddy et al., 2021a) for different joint neural prior training objectives

| Model | PESQ (↑) | SIG (↑) | BAK (↑) | OVRL (↑) | UTMOS (↑) | WV-MOS (↑) | NISQA (↑) |
|---|---|---|---|---|---|---|---|
| Voicefixer-GSR (Liu et al., 2022) | | | | | | | |
| KL-only | 2.08 | 3.15 | 3.91 | 3.13 | 3.12 | 3.37 | 3.68 |
| JNP | **2.15** | **3.43** | **4.09** | **3.20** | **3.58** | **3.86** | **3.76** |
| DNS-Real (Reddy et al., 2021a) | | | | | | | |
| Model | | SIG (↑) | BAK (↑) | OVRL (↑) | UTMOS (↑) | WV-MOS (↑) | NISQA (↑) |
| KL-only | | 3.03 | 3.88 | 2.97 | 2.75 | 3.31 | 4.16 |
| JNP | | **3.39** | **4.03** | **3.12** | **3.15** | **3.69** | **4.34** |

an ablation study with two bridge models trained with the multi-measure aligned joint neural prior and KL-aligned-only prior respectively, to prove that our multi-measure alignment is necessary. The results are shown in Table 19. Our joint neural prior clearly outperforms the KL-only alignment proposed in Fang et al. (2021), proving our method's effectiveness.

Table 20: Evaluation results on curated testsets with different degradation types for bridge models with and without joint neural prior.

| Model | PESQ (↑) | DNSMOS (↑) | WVMOS (↑) | UTMOS (↑) | NISQA (↑) |
|---|---|---|---|---|---|
| Effect of JNP on Perceptual Metrics | | | | | |
| Noise | | | | | |
| w/ JNP | **1.88** | **3.17** | **4.05** | **3.69** | **4.36** |
| w/o JNP | 1.75 | 2.95 | 4.01 | 3.65 | 4.10 |
| Reverb | | | | | |
| w/ JNP | **2.12** | **3.24** | **4.11** | **3.77** | **4.60** |
| w/o JNP | 2.01 | 3.13 | 4.10 | 3.72 | 4.43 |
| BWE | | | | | |
| w/ JNP | **2.70** | **3.30** | **4.32** | **3.97** | **4.69** |
| w/o JNP | 2.51 | 3.19 | 4.26 | 3.95 | 4.50 |
| CLIP | | | | | |
| w/ JNP | **3.26** | **3.25** | **4.31** | **4.01** | **4.66** |
| w/o JNP | 2.95 | 3.16 | 4.20 | 3.91 | 4.63 |

### I.4 ABLATION ON JOINT NEURAL PRIOR'S EFFECTIVENESS FOR DIFFERENT DEGRADATION TYPES

To further discuss whether the joint neural prior can converge the prior of diverse degradation types to a unified clean distribution, rather than only working on certain types, we conduct an ablation study on the effectiveness of joint neural prior for different degradation types. We train two bridge models with and without joint neural prior respectively, and evaluate their performance on different degradation types on curated testsets with additional noise, reverberation, bandwidth limitation, and clipping on the VCTK corpus. The models are trained for 500k steps. Results are shown in Table 20. The results demonstrate consistent improvement across all dereverberation types, indicating that JNP benefit all distortions simultaneously.

