# OpenReview forum: "VoiceBridge: Designing Latent Bridge Models for General Speech Restoration at Scale"
_ICLR.cc/2026/Conference — Submitted to ICLR 2026_

### Official Review · Reviewer_mn36 · 2025-10-29

**Soundness:** 3
**Presentation:** 3
**Contribution:** 3
**Rating:** 6
**Confidence:** 4

**Summary:**

The paper proposes using a latent-space bridge model for the task of speech enhancement. The bridge is trained between the latent representations of clean and degraded speech recordings, and the latent space is constructed using VAE. They authors propose several techniques to make VAE more suitable for encoding waveforms. Overall, the paper is quite solid and well-written.

**Strengths:**

The paper has the following key strengths:

1. Paper is well-structured and easy to read. The authors provide a detailed description of the training setup, and the training procedure is presented clearly.
2. The authors propose a set of novel techniques that enhance the training of VAEs for constructing latent audio representations. They also provide an ablation study to validate the importance of each design choice.
3. The authors show that the proposed method performs very well on several benchmarks for speech enhancement, as well as on other tasks (codec artefact removal, denoising)

**Weaknesses:**

However, the following weaknesses can be identified:

1. Some relevant speech enhancement methods, though cited in the paper, are not included in the benchmark comparisons. For instance, the authors could consider adding FINALLY [1], the popular HiFi-GAN-2 [2] baseline, and other methods like MIIPHER [3] and Genhanser [4] to the comparisons.
2. In many speech enhancement applications, inference speed plays a crucial role. However, bridge methods typically require multiple steps to obtain the final samples. It would be interesting to see the inference time comparison between the presented model and the other benchmarks.

**Questions:**

I have the following suggestions for the authors to consider:

- Since the presented model supposedly solves the Schrödinger bridge problem, it would be interesting to compute Path KL, which is the estimate of how good the model solves optimal transport.
- Moreover, other papers, e.g. FINALLY, claim that speech enhancement necessitates finding such an enhanced audio $\hat{x}_0$ that is the closest (in some meaningful sense) to the original noisy audio $x_1$. It would also be interesting to compare the baselines and the presented model regarding "how far" the enhanced audio is from the noisy input.


#### **References:**

[1] Babaev et al., "FINALLY: fast and universal speech enhancement with studio-like quality"

[2] Su et al., "HiFi-GAN-2: studio-quality speech enhancement via generative"

[3] Koizumi et al., "MIIPHER: a robust speech restoration model integrating self-supervised speech and text representations"

[4] Yang et al., "Genhancer: high-fidelity speech enhancement via generative modeling on discrete codec tokens"

---

> ### Author Response · Authors · 2025-11-21
> **Response to Reviewer mn36, part 1**
>
> Thank you for the positive and constructive review, and for noting the clarity of the training setup and the strength of our VAE ablations and benchmark performance. We address your comments and suggestions below.
>
> ***
>
> ## 1. Missing baselines: FINALLY, HiFi-GAN-2, MIIPHER, Genhancer
>
>
> > *“The authors could consider adding FINALLY, HiFi-GAN-2, MIIPHER, Genhanser to the comparisons.”*
>
> We agree these are important recent methods and appreciate you listing them explicitly.
>
> - **Current baseline set.**  Our current experiments already include **strong universal baselines**, such as Resemble-Enhance, UniverSE++, AnyEnhance, and VoiceFixer, which mainly focus on diverse SE/GSR tasks. VoiceBridge consistently matches or surpasses these on both **objective** and **perceptual / human MOS** metrics.
>
> - **What we will add.**  We fully agree that the paper should more explicitly discuss these methods:
> 	- For Miipher, we will add corresponding results onto our evaluation benchmarks. We evaluate Miipher on VoiceFixer-GSR and DNS-Real datasets using open-source code, and compare it with VoiceBridge:
>
>     - On VoiceFixer-GSR:
>
>     | Model | PESQ | SIG | BAK | OVRL | UTMOS | WV-MOS | NISQA |
>     |------------|------|--------|-------|-------|-------|--------|--------|
>     | Miipher    | 1.491 | 3.335 | 3.992 | 3.051 | 3.617 | 4.171 | 4.136  |
>     | VoiceBridge| 2.436 | 3.469 | 4.083 | 3.210 | 4.330 | 4.246 | 4.098 |
>
>     - On DNS-Real:
>
>     | Model | SIG | BAK | OVRL | UTMOS | WV-MOS | NISQA |
>     |-----|------|--------|-------|-------|-------|--------|
>     | Miipher    | 3.325 | 3.976 | 3.171 | 3.911 | 3.171 | 4.124 |
>     | VoiceBridge|  3.447 | 3.985 | 3.166 | 4.197 | 4.034 | 4.497 |
>
> 	- Hifi-GAN-2, FINALLY and Genhancer are closed-source models, which we cannot easily evaluate on our benchmarks. We will further include a section that compares VoiceBridge with the reported results from the corresponding papers.
>
> We believe incorporating these discussions and any feasible evaluations will make our positioning w.r.t. recent universal SE models clearer.

---

> > ### Comment · Reviewer_mn36 · 2025-11-24
> >
> > I would like to thank the authors for their responses.
> >
> > - A couple of notes on the HiFi-GAN2 and FINALLY. As I know, FINALLY has the links to the validation datasets on their webpage: https://mmacosha.github.io/finally-demo/
> > - HiFi-GAN-2 model has also released audio samples cleaned using their model: https://daps.cs.princeton.edu/projects/Su2021HiFi2/index.php?env-pairs=DAPS&speaker=f10&src-env=all
> > (although I would agree that the number of samples is small in this case)

---

> > > ### Comment · Reviewer_mn36 · 2025-11-26
> > >
> > > I want to thank the authors for their responses. I would be happy if the comparisons with the models mentioned above were included in the final version of the manuscript. Overall, I am satisfied with the responses and maintain my score.

---

> > > ### Author Response · Authors · 2025-11-27
> > > **Response to Official Comment by Reviewer mn36**
> > >
> > > Thank you for your response! We evaluated VoiceBridge on the open samples from the "Additional Comparsion with HiFi-GAN-2" section in <https://mmacosha.github.io/finally-demo/>. The results are as follows:
> > >
> > > | Model | DNSMOS | UTMOS | WV-MOS |
> > > |------------|------|--------|-------|
> > > | HiFi-GAN-2 | 3.283 | 3.947 | 3.470 |
> > > | FINALLY | **3.313** | 3.847 | 3.989 |
> > > | VoiceBridge| 3.063 | **4.111** | **4.034** |
> > >
> > > Note that VoiceBridge achieves on-par performance with FINALLY, though the latter is pretrained on a large-scale internal dataset, while VoiceBridge is trained only on publicly available data. Meanwhile, HIFI-GAN-2 and FINALLY are both point-to-point matching moodels, while VoiceBridge models distributions, which may be more generalizable to diverse unseen degradations.

---

> ### Author Response · Authors · 2025-11-21
> **Response to Reviewer mn36, part 2**
>
> ## 2. Inference speed and number of steps
>
>
> > *“Bridge methods typically require multiple steps… It would be interesting to see inference time comparison between the presented model and other benchmarks.”*
>
> This is a very valuable point, especially for real-world SE applications. In **Appendix H, Table 14**, we list the Real-Time-Factor (RTF) of VoiceBridge and baseline models, demonstrating that VoiceBridge achieves faster inference than other diffusion / flow-based baselines. Although it is still slower than GAN-based models such as VoiceFixer with just one NFE, it is fast enough for real-time speech enhancement. This is achieved by:
>
> - **Architectural efficiency.**  While bridge models are iterative, our design deliberately mitigates this cost by:
> 	- Operating entirely in a **compressed latent space** (via the VAE), dramatically shortening sequence length compared to raw waveforms / spectrograms.
>
> 	- Using a **small, fixed number of bridge steps** (we do *not* use tens or hundreds of diffusion steps typical of classic diffusion models).
>
>
> We expect this will directly address your concern and provide a more complete picture of practicality.

---

> ### Author Response · Authors · 2025-11-21
> **Response to Reviewer mn36, part 3**
>
> ## 3. Path KL as a measure of how well the SB is solved
>
>
> > *“Since the presented model supposedly solves the Schrödinger bridge problem, it would be interesting to compute Path KL.”*
>
> This is an excellent question! We agree that **Path KL** is a conceptually appropriate diagnostic for SB models and an interesting way to evaluate how well the bridge approximates the optimal transport.
> As a measure for SB model convergence, we may calculate the KL divergence between the actually solved path and the reference stochastic process, illustrated in Equation 9.
>
> Concretely, consider:
>
> * Reference process $P_\text{ref}$:
>   $$
>   dX_t = f^{\text{ref}}(t,X_t)dt + \sqrt{2\varepsilon}dW_t,
>   $$
> * Learned (controlled) process $P_\theta$:
>   $$
>   dX_t = f_\theta(t,X_t)dt + \sqrt{2\varepsilon}dW_t,
>   $$
>   with same diffusion coefficient (\sqrt{2\varepsilon}).
>
> By Girsanov, the **KL between path measures** $P_\theta$ and $P_\text{ref}$ on trajectories ${X_t}, {t\in[0,1]}$ is
>
> $$
> \mathrm{KL}\big(P\_\theta \Vert P\_\text{ref}\big)=\frac{1}{4\varepsilon} \mathbb{E}_{P\_\theta}\left[\int_0^1 \big| f\_\theta (t,X\_t) - f^{\text{ref}}(t,X\_t)\big|^2dt \right].
> $$
>
> In actual training, the trajectories are discretized into timesteps, and the KL divergence can be approximated using the Riemann sum of the norm of the delta-drift term.
>
> $$
> \mathrm{KL}\big(P\_\theta \Vert P\_\text{ref}\big)\propto
> \mathbb{E}\_{P\_\theta}\left[
> \sum\_{t=0}^T \big|f\_\theta(t,X\_t) - f^{\text{ref}}(t,X\_t)\big|^2dt
> \right].
> $$
>
> This term can be then reparameterized into the diffusion loss term in training. In fact, the KL divergence can be basically viewed as an expectation of training losses over different diffusion timesteps. The exact number highly depends on the latent space and diffusion coefficients.

---

> ### Author Response · Authors · 2025-11-21
> **Response to Reviewer mn36, part 4**
>
> ## 4. “How far” the enhanced audio is from the noisy input
>
>
> > *“Other papers (e.g. FINALLY) claim SE necessitates finding an enhanced ( \\hat{x}\_0 ) that is closest to the noisy audio ( x\_1 ). It would be interesting to compare baselines and the presented model regarding ‘how far’ the enhanced audio is from the noisy input.”*
>
> This is a great suggestion and touches on a subtle but important point:
>
> - In **classical denoising**, minimal change from (x\_1) is indeed desirable: you want to remove only the noise and keep everything else untouched.
>
> - In **general speech restoration**, some tasks (e.g., **bandwidth extension, heavy reverberation, codec artifacts, TTS artifacts**) inherently require *larger* changes from (x\_1). For instance, reconstructing high frequencies or de-reverberating a long tail will move the waveform further away from the original noisy/low-quality signal.
>
> - FINALLY claims that the SE task can be solved by only modeling $\text{argmax} p\_\\theta(y|x)$, not $ p\_\\theta(y|x)$, and thus a **GAN model** instead of diffusion is sufficient. However, in the general, severe scenarios such as **heavy noise/reverberation, exceptionally low bandwidth**, we believe that $p\_\\theta(y|x)$ should be highly complex and multimodal, and only modeling the $\text{argmax}  p\_\\theta(y|x)$ using regression creates **less fine-grained results** than actually sampling on $p\_\\theta(y|x)$ using generative models. In fact, FINALLY actually proposes **the "oversmoothing" problem of their predicted speech** (and thus utilizes SSL features to achieve disentanglement), which might be the **direct result of their regression modeling**.
>
>
> Our SB formulation and JNP are designed to produce **smooth, regular transport** in latent space between LQ and HQ distributions. However, the nature of transport models and iterative sampling make the target and prior distributions more distinct and distant.
>
> We fully agree that quantifying this is informative. To quantify "how far", we calculated the $L^2$ distance of Wav2Vec2[1] features between the input and enhanced audio, averaged over the DNS-Real test set:
>
> | Model        | Wav2Vec2 Feature Distance |
> |--------------|---------------------------|
> | VoiceFixer   | 2.199 $\pm$ 0.897         |
> | UniverSE++   | 2.010 $\pm$ 0.750         |
> | Resemble-Enhance  | 2.163 $\pm$ 0.870    |
> | VoiceBridge  | 2.207 $\pm$ 0.844         |
> | Miipher     | 2.133 $\pm$ 0.712          |
>
> VoiceBridge results in the maximum distance between input and enhanced audio. However, rather than implying that **"the restored audio went too far from the original"**, it might instead be suggesting that VoiceBridge **restores the degradations of the input most completely**. What these distance truly imply is hard to distinguish and deserves further investigation.
>
> ***
>
> Once again, thank you for your supportive and thoughtful review, and for the concrete suggestions (additional baselines, runtime, Path KL, and distance-to-input analysis). We will incorporate these in the revision to make the paper’s empirical and methodological contributions clearer and more complete.
>
> [1] Baevski A, Zhou Y, Mohamed A, et al. wav2vec 2.0: A framework for self-supervised learning of speech representations[J]. Advances in neural information processing systems

---

> ### Author Response · Authors · 2025-11-27
> **Response to Official Comment by Reviewer mn36**
>
> Thank you very much for your time and constructive feedback, which has helped us a lot in improving the manuscript.

---

### Official Review · Reviewer_DuQh · 2025-10-29

**Soundness:** 4
**Presentation:** 2
**Contribution:** 2
**Rating:** 4
**Confidence:** 4

**Summary:**

This paper introduces a latent bridge model (LBM) for speech restoration and enhancement. The authors introduce an extensive training scheme for a variational autoencoder (VAE) in order to have both an energy preserving latent space as well as a joint prior for both high-quality and low-quality samples which have been arbitrarily degraded from the same sample. The authors show that this makes the latent diffusion easier to solve leading to an improved bridge model down the line. The LBM itself is a transformer with 544M parameters. In the end, the authors also introduce a joint fine-tuning stage for the bridge model and the decoder to generate perceptually better samples using human feedback and a descriminator-based adversarial and feature-matching loss.

**Strengths:**

- The proposed system was explained in detail and in a very concise and understandable way.
- The experiments and ablations support the claim that their approaches improve their own baseline system.

**Weaknesses:**

- The overall novelty of the proposed improvements is low:
	- Learning a joint neural prior in a VAE which is used in an LQ to HQ speech enhancement/restoration system is not novel (see Fang et al., "Variational Autoencoder for Speech Enhancement with a Noise-Aware Encoder", ICASSP 2021)
	- Ensuring the energy level stays consistent in the latent space is not a novel idea in itself. While the specific learning target using a joint scaling in the loss formulation might be novel, Karras et al. "Analyzing and Improving the Training Dynamics of Diffusion Models", CVPR 2024, already showed that diffusion models benefit from magnitude preserving layers in the neural network architecture. More recently, Richter et al., "Normalize Everything: A Preconditioned Magnitude-Preserving Architecture for Diffusion-Based Speech Enhancement", WASPAA 2025, showed that this also holds for Schrödinger-bridge-based speech enhancement models. However, we acknowledge that the latter was published within the last two months and was previously only available on arXiv.
	- LBMs and transformer based speech enhancement models using diffusion, Schrödinger bridges or flow matching are not novel as well.
- There was no comparison with recent universal speech enhancement models, namely
	- Ku et al. "Generative Speech Foundation Model Pretraining for High-Quality Speech Extraction and Restoration", ICASSP 2025
	  This work is cited. However, no comparison is done with regards to their results. This is especially important as their results on the speech denoising task indicate significantly higher performance when compared to the same baselines.
	- Liu et al., "Generative Pre-Training for Speech with Flow Matching", ICLR 2024
	  This work was not cited but the same holds for the results here, how does the proposed approach compare to SpeechFlow?

**Questions:**

- How does the model compare to the recent SOTA approaches:
	- Ku et al. "Generative Speech Foundation Model Pretraining for High-Quality Speech Extraction and Restoration", ICASSP 2025
	- Liu et al., "Generative Pre-Training for Speech with Flow Matching", ICLR 2024

- The clean speech VAE reconstructed, i.e., $\hat{\bf{x}}_0=\mathcal{D}(\mathcal{E}^{\text{np}}(\bf{x}_0))$, metrics were reported. However, the degraded and then only using the VAE as enhancement module output was not investigated. As the joint neural prior should make it possible to reconstruct the clean sample in the best case, the metrics for $\hat{\bf{x}}_0=\mathcal{D}(\mathcal{E}^{\text{np}}(\bf{x}_1))$ would be interesting to report.

- Please consider renaming the fine-tuning loss which should optimize for perceptual quality as the term "human feedback" implies something differently. From our understanding the $\mathcal{L}_\text{hf}$ loss is a perceptual loss based on PESQ and UTMOS. Thus, it should be named accordingly.

---

> ### Author Response · Authors · 2025-11-21
> **Response to Reviewer DuQh, part 1**
>
> Thank you for the detailed and knowledgeable review, and for acknowledging the clarity of the system description and the strength of the ablations. We address your concerns point-by-point.
>
> ***
>
> ## 1. Novelty of the joint neural prior vs. prior “noise-aware” VAEs
>
>
> > *“Learning a joint neural prior in a VAE which is used in an LQ to HQ speech enhancement/restoration system is not novel (see Fang et al., 2021).”*
>
> We appreciate you pointing out Fang et al., *“Variational Autoencoder for Speech Enhancement with a Noise-Aware Encoder”* (ICASSP 2021). Fang et al. pioneered the idea of using noisy–clean pairs to train a *noise-aware encoder* whose latent posterior approximates that of a clean-speech VAE, inside a VAE-NMF denoising framework. However, our joint neural prior (JNP) differs from Fang et al. in multiple aspects:
>
> - Our JNP is trained using multiple alignment objectives. As illustrated in Equation 6, we align the clean and degraded latents not only using a KL loss (used by Fang et al., which is equal to the MSE term in L\_latent in Equation 6), but also cosine similarity in latent space and multiple supervision signals in the reconstructed data space, in order to reduce the distance between the latents in all geometrical measures. Meanwhile, we explicitly included the EP training term within the JNP training process, to ensure the energy preservation during JNP training.
>
> - To prove that such all-measure alignment is necessary, we conduct an ablation study where only the KL loss by Fang et.al. is included, as shown below:
>
> | Model          | PESQ  | SIG | BAK | OVRL | UTMOS | WV-MOS | NISQA |
> |----------------|-------|-----|-----|------|-------|--------|-------|
> | KL-only  | 2.08 | 3.15 | 3.91 | 3.13 | 3.12 | 3.37 | 3.68 |
> | JNP | **2.15** | **3.43** | **4.09** | **3.20** | **3.58** | **3.86** | **3.76** |
>
> On DNS-Real:
>
> | Model          | SIG | BAK | OVRL | UTMOS | WV-MOS | NISQA |
> |----------------|-----|-----|------|-------|--------|-------|
> | KL-only  | 3.03 | 3.88 | 2.97 | 2.75  | 3.31 | 4.16 |
> | JNP | **3.39** | **4.03** | **3.12** | **3.15** | **3.69** | **4.34** |
>
>
>
> - Our **Joint Neural Prior (JNP)** generalizes this idea to a **multi-distortion, waveform-level, energy-preserving latent space**, and uses it for a **latent Schrödinger Bridge** that performs **general 48 kHz speech restoration** (noise, reverb, BWE, clipping, codec, TTS) in **one unified model**.
>
> - Technically, the novelty is in:
> 	- Introducing the multi-measure alignment which involves latent space MSE (KL) and cosine similarity feature loss along with the data-space loss, and proving its effectiveness.
>
> 	- Designing **EP-VAE + JNP** so that **all degradation priors are contracted toward a common, energy-preserving HQ manifold**,
>
> 	- Integrating this with a **latent SB** that leverages the contracted joint prior to efficiently learn a data-to-data bridge across many distortions,
>
> 	- Demonstrating this in a **general GSR setting** far beyond additive noise denoising.
>
>
> We will revise the related work section to explicitly discuss Fang et al., clearly acknowledge the conceptual similarity (latent prior refinement for SE), and then spell out these differences in objective, supervision and multi-distortion scope.

---

> > ### Comment · Reviewer_DuQh · 2025-11-25
> > **Reviewer's Response to Rebuttal Part 1**
> >
> > We thank the authors for their time and for addressing the highlighted weaknesses and questions.
> >
> > We very much appreciate the additional ablation presented using the KL-only training. Including the work of Fang et al. into the related work and showing the improvements the JNP achieves over it also in the paper or appendix will strengthen the claim for novelty in our view.

---

> ### Author Response · Authors · 2025-11-21
> **Response to Reviewer DuQh, part 2**
>
> ## 2. Energy-preserving latent space vs. magnitude-preserving architectures
>
>
> > *“Ensuring the energy level stays consistent in the latent space is not a novel idea in itself (Karras et al. 2024; Richter et al. 2025).”*
>
> You are absolutely right that **magnitude preservation wihtin neural network layers** has recently attracted a lot of attention. Both Karras et al. and Richter et al. proposed magnitude-preserving layers to eliminate magnitude drift and enhance generation quality. **However, our EP-VAE is inspired by other underlying principles and has a completely different mechanism.**
> Karras et al. and Richter et al. both forces the activation on each channel to remain at the same magnitude level by dividing the weight norm channel-wise. This is a **network-architecture design** to stabilize the network output across layers.
> Our EP-VAE, however, is a **training strategy design,** without adding additional constraints to the network's forward process. We are not ensuring a constant energy level **within the activations**, but an equivariance between **data and latent spaces**. The input of the network still varies in energy levels, but a direct relationship between latent energy and waveform energy is enforced, via the loss in Eq. (5) (joint amplitude scaling plus energy-matching).
> This is applied **inside the VAE** that defines the latent space where the SB operates, not inside the SB network itself. In contrast Karras et al. and Richter et al. adjust diffusion architectures in data or latent image space.
> In other words, prior works normalize/condition the **denoiser/SB network’s internal activations**, whereas we **explicitly regularize the latent representation of speech itself** so that:
>
> 1. HQ and LQ signals share a well-structured, energy-consistent latent manifold;
>
> 2. The JNP can meaningfully contract LQ distributions toward HQ;
>
> 3. The SB learns smoother bridges in this latent space.

---

> > ### Comment · Reviewer_DuQh · 2025-11-25
> > **Reviewer Response to Rebuttal Part 2**
> >
> > While we agree that the main benefit of energy preservation throughout the network is improved stability, we disagree that the stated goal of the EP-VAE is fundamentally different. If a network will keep the magnitude stable throughout its layers this will ultimately also lead to an equivariance between input and output (here: latents) of a network. Thus, treating encoder and decoder of the VAE as individual entities will lead to the same equivariance. This is regardless of whether one has skip connections as in the U-Net from Karras et al. or without skip connections between encoder and decoder as in your VAE.
> > We acknowledge that a loss-driven design as you have proposed may have advantages, however, neither do you adequately show these advantages nor are we aware of any literature which shows this.

---

> > > ### Author Response · Authors · 2025-11-28
> > > **Response to Reviewer DuQh**
> > >
> > > Thank you! We sincerely appreciate your pointing out that maintaining a stable magnitude through the neural network layers leads to similar effects as in our EP-VAE. This is a remarkable observation. We agree that the two methods share a common goal of preserving signal energy.
> > >
> > > However, we respectfully clarify that the motivations and implementations are different. Our EP-VAE design aims to create a latent space which is energy-preserving with the waveform domain, so that the Schrodinger Bridge in this space inherits the good properties of those in data space. The core motivation lies in **building a better-structured latent space**, rather than **stabilizing training dynamics**. We are not aware of prior works that explicitly enforce energy preservation in latent space for Schrodinger Bridge modeling.
> > >
> > > Besides, EP-VAE is only one of our introduced novelties. We further propose other elements such as the joint neural prior (JNP) which can be integrated to leverage the advantages associated with EP-VAE.
> > >
> > > We are very grateful for your insightful comments, which help us better position our contributions. We will cite and discuss the mentioned work in the revised version to clarify the similarities and differences.

---

> ### Author Response · Authors · 2025-11-21
> **Response to Reviewer DuQh, part 3**
>
> ## 3. "LBMs / transformer SE not being novel"
>
>
> > *“LBMs and transformer based speech enhancement models using diffusion, Schrödinger bridges or flow matching are not novel as well.”*
>
> We agree and do not claim that “LBM” or “transformer for SE” is itself novel. The novelty is in the **combination and setting**:
>
> - **Setting:**  a **single latent SB model for 48 kHz general speech restoration** across many distortions (noise, reverb, BWE, clipping, codec artifacts, TTS artifacts), rather than a single-task denoiser or dereverber.
>
> - **Design:**  EP-VAE + JNP explicitly tailored to make **multi-distortion latent SB tractable and stable**, plus a perceptual fine-tuning regime tightly integrated with this backbone.
>
> - **Scope of experiments:**  including **zero-shot codec-artifact removal** and **TTS post-processing** with strong perceptual gains, which (to our knowledge) prior SB or flow-matching SE works have not systematically evaluated.
>
>
> We will tighten the contribution section to reflect this more clearly and to avoid any implication that the basic ingredients (SB, transformers, VAEs) are themselves new.

---

> > ### Comment · Reviewer_DuQh · 2025-11-25
> > **Reviewer Response to Rebuttal Part 3**
> >
> > We appreciate that the authors want to clarify and consolidate their contributions such that it is easier for a reader grasp the novelty correctly.

---

> ### Author Response · Authors · 2025-11-21
> **Response to Reviewer DuQh, part 4**
>
> ## 4. Comparison to SpeechFlow and Ku et al. (2024/2025)
>
>
> > *“There was no comparison with recent universal speech enhancement models … Ku et al., ICASSP 2025; Liu et al., ICLR 2024 (SpeechFlow). How does the model compare?”*
>
> Thank you for flagging this — these are fascinating, highly relevant works, demonstrating the significance of generative pretraining on downstream low-level tasks.
>
> - **SpeechFlow (Liu et al., ICLR 2024)** pretrains a **flow-matching generative model** on **60k hours of untranscribed speech**, then fine-tunes for various tasks (enhancement, separation, TTS, etc.).
>
> - **Ku et al. 2024/2025** shift the SpeechFLow training space from mel spectrogram to complex STFT avoiding the limitations brought by vocoders.
>
>
> However, we did not directly compare VoiceBridge with these two works because:
>
> 1. **Domain differences**
> 	1. Both SpeechFlow and Ku et al. work at a 16kHz sampling rate, while ours directly generates 48kHz audio. This difference brings a significant gap in modeling difficulties, and makes direct comparison unfair, especially since most evaluation protocols only use the lower 16kHz part to calculate a metric.
>
> 	2. Both SpeechFlow and Ku et al. need a specific finetuning process for each restoration task, which means they are not general models that solve all tasks with a single model. Instead, different sets of parameters are used to restore different distortion types, which makes the generation process much easier as it does not require the model to handle diverse degraded priors.
>
> 2. **Pretraining scale**
> 	1. SpeechFlow and Ku et al. both use an extra-large scale pre-training dataset (Libri Light, 60k hours), far larger than our training set (1138 hours). Datasets at that scale are not even publicly available for 48kHz clean speech.
>
> 	2. VoiceBridge is trained on a **smaller, curated multi-degradation dataset**. Our focus is on **how to structure the latent space and bridge** for GSR when you *do not* have massive foundation-model pretraining.
>
> 3. **Task coverage mismatch**
> 	1. SpeechFlow & Ku et al. report impressive results on denoising and several tasks, but **TTS artifact post-processing** and some of our GSR settings (e.g., combining harsh reverb with other distortions, podcast-style TTS) are, to the best of our knowledge, not studied there. There is also a mismatch between the specific benchmark and metric we select.
>
> 	2. SpeechFlow & Ku et al. are not open-sourced, which means we cannot evaluate them in our test settings.
>
>
> [1] Ku et al. "Generative Speech Foundation Model Pretraining for High-Quality Speech Extraction and Restoration", ICASSP 2025
>
> [2] Liu et al., "Generative Pre-Training for Speech with Flow Matching", ICLR 2024

---

> > ### Comment · Reviewer_DuQh · 2025-11-25
> > **Reviewer Response to Rebuttal Part 4**
> >
> > We acknowledge that 16kHz and 48kHz constitute different problem settings. However, a downsampled audio generated using a 48kHz model can be directly compared with a 16kHz model making a direct comparison possible.

---

> > > ### Author Response · Authors · 2025-11-28
> > > **Response to Reviewer DuQh**
> > >
> > > Thank you very much for your constructive response. We further compared VoiceBridge with SpeechFlow and Ku et al.'s method on the same evaluation benchmarks, downsampling our outputs to 16kHz to match their settings. The results are as follows:
> > >
> > > 1. On VB-Demand:
> > >
> > >     | Model | PESQ | WVMOS | ESTOI | SISDR |
> > >     |------------|------|--------|-------|-------|
> > >     | SpeechFlow | 3.13 | - | 0.87 | - |
> > >     | Ku et al.  | 3.27 | 4.41 | 0.88 | 19.1 |
> > >     | VoiceBridge| 2.82 | 4.39 | 0.79 | 4.8 |
> > >
> > > 2. On WSJ0-CHiME3:
> > >
> > >     | Model | PESQ | WVMOS | ESTOI | SISDR |
> > >     |------------|------|--------|-------|-------|
> > >     | Ku et al.  | 2.85 | 4.26 | 0.92 | 16.2 |
> > >     | VoiceBridge| 1.72 | 4.36 | 0.82 | 3.4 |
> > >
> > > 3. On WSJ0-BWE
> > >
> > >    | Model | WVMOS | SQUIM-MOS | ESTOI | SISDR |
> > >    |------------|------|--------|-------|-------|
> > >    | Ku et al.  | 3.93 | 4.41 | 0.92 | 8.5 |
> > >    | VoiceBridge| 4.28 | 4.48 | 0.85 | 3.5 |
> > >
> > > We must state that:
> > >
> > > 1. SpeechFlow and Ku et al.'s methods need specific finetuning for each task, while VoiceBridge is a general model without any task-specific adaptation.
> > > 2. On waveform-level metrics like SISDR, latent-space methods like VoiceBridge typically underperforms data-level methods due to the latent reconstrucion error **(Especially, VAE reconstruction is trained mainly on MR-STFT maginitude, not on the waveform dimension, and no phase preservation is guaranteed)** and the cascading error from latent modeling. However, these metrics do not necessarily reflect the generated audio quality, as pointed out in [1,2,3].
> > >
> > > [1] Manocha, P., Jin, Z., and Finkelstein, A. Audio similarity is unreliable as a proxy for audio quality, 2022.
> > >
> > > [2] Manjunath, T. Limitations of perceptual evaluation of speech quality on voip systems.
> > >
> > > [3] Andreev, P., Alanov, A., Ivanov, O., and Vetrov, D. Hifi++: a unified framework for bandwidth extension and speech enhancement.

---

> ### Author Response · Authors · 2025-11-21
> **Response to Reviewer DuQh, part 5**
>
> ## 5. VAE-only enhancement: ( $\hat{x}_0 = \mathcal{D}(\mathcal{E}^\text{np}(x_1))$ )
>
>
> > *“The clean speech VAE reconstructed metrics were reported … however, the degraded and then only using the VAE as enhancement module output was not investigated.”*
>
> We agree that this is a very informative diagnostic: it directly measures **how far the joint neural prior alone can go**, without the LBM. Also,
>
> - Conceptually, JNP is designed so that $\mathcal{E}^\text{np}(x_1)$ **lies close to** $\mathcal{E}(x_0)$ in latent space when $x_1$ is a degraded version of $x_0$, in measures both within the latent space and the decoded data space.
>
> - Therefore, $\hat{x}_0 = \mathcal{D}(\mathcal{E}^\text{np}(x_1))$ should already yield a **non-trivial enhancement** performance, but we expect the **LBM to further refine** details and perceptual quality, especially for more structured distortions (reverb, codec artifacts, TTS artifacts).
>
> We evaluated the performance of VAE-only enhancement:
>
> | Model | PESQ | DNSMOS | WVMOS | NISQA |
> | -- | -- | -- | -- | -- |
> | LQ | 1.97 | 2.69 | 2.99 | 3.00 |
> | VAE-only | 2.31 | 2.94 | 3.38 | 3.71 |
> | SB | 2.82 | 3.20 | 4.39 | 4.51 |
> | HQ | - | 4.22 | 4.5 | 4.62 |
>
> This indicates that VAE indeed performs non-trivial restoration due to the JNP alignment. However, core aspects of the restoration process still rely on the LBM for optimal performance.

---

> > ### Comment · Reviewer_DuQh · 2025-11-25
> > **Reviewer Response to Rebuttal Part 5**
> >
> > We thank the authors for this additional ablation and we think this would be a good addition in the appendix. Possibly even distinguishing between different degradation types. This could further demonstrate in which cases the SB model needs to do the heavy lifting.

---

> > > ### Author Response · Authors · 2025-11-28
> > > **Response to Reviewer DuQh**
> > >
> > > Thank you for your recognition of our added ablations. To further investigate which case the SB model needs heavy-lifting, we conducted additional experiments on different degradation types in curated evaluation sets. The results are as follows:
> > >
> > > | Noise  | PESQ | DNSMOS | WVMOS | UTMOS | NISQA |
> > > |--------|------|--------|-------|-------|-------|
> > > | VoiceBridge | 2.34 | 3.17 | 4.30 | 4.26 | 4.18 |
> > > | VAE | 1.71 | 2.58 | 3.51 | 3.37 | 3.23 |
> > >
> > > | Reverb | PESQ | DNSMOS | WVMOS | UTMOS | NISQA |
> > > |--------|------|--------|-------|-------|-------|
> > > | VoiceBridge | 2.67 | 3.24 | 4.41 | 4.31 | 4.63 |
> > > | VAE | 1.52 | 2.72 | 3.85 | 3.69 | 4.20 |
> > >
> > > | BWE | PESQ | DNSMOS | WVMOS | UTMOS | NISQA |
> > > |--------|------|--------|-------|-------|-------|
> > > | VoiceBridge | 3.32 | 3.23 | 4.41 | 4.31 | 4.64 |
> > > | VAE | 2.63 | 2.83 | 3.75 | 3.79 | 4.41 |
> > >
> > > | CLIP | PESQ | DNSMOS | WVMOS | UTMOS | NISQA |
> > > |--------|------|--------|-------|-------|-------|
> > > | VoiceBridge | 3.57 | 3.23 | 4.43 | 4.31 | 4.69 |
> > > | VAE | 2.94 | 2.82 | 3.96 | 3.90 | 4.44 |
> > >
> > > Basically, we can find that VAE-only enhancement works well on Bandwidth extension task and de-clipping tasks. These two tasks are considered easier, since they preserve the main structure of the speech signal. The Bridge model successfully refines all degradations to a similar HQ leel beyond the VAE reconstruction. We will add the above ablation to the Appendix in the revised version.

---

> ### Author Response · Authors · 2025-11-21
> **Response to Reviewer DuQh, part 6**
>
> ## 6. Naming of the “human feedback” loss
>
>
> > *“Please consider renaming the fine-tuning loss … as the term ‘human feedback’ implies something differently. From our understanding the $\mathcal{L}_\text{hf}$ loss is a perceptual loss based on PESQ and UTMOS.”*
>
> You are absolutely right. The current name is indeed a little misleading:
>
> - The loss is computed using PESQ and UTMOS predictors, where UTMOS itself was trained on human MOS data, but we do not collect online human feedback or perform RLHF-style optimization.
>
> - A better name would be something like “PESQ/UTMOS-based perceptual loss” or “perceptual loss” for simplicity.
>
>
> We will rename it into $\mathcal{L}_\text{perc}$ or similar, and clearly state that this uses *frozen* metric predictors trained on human ratings, not live human feedback.
>
> ***
>
> Once again, thank you for the constructive feedback and the concrete suggestions (especially the VAE-only ablation and the naming of the perceptual loss). We will incorporate these changes to better position the work and sharpen the main messages.

---

> > ### Comment · Reviewer_DuQh · 2025-11-25
> > **Reviewer Response to Rebuttal Part 6**
> >
> > We thank the authors for renaming the loss term accordingly and look forward to a more descriptive presentation.
> >
> > ### General Remarks
> >
> > We further observed that few metrics were reported which do not resort to audio quality alone. You report on WER for the TTS enhancement task. However some of the other tasks could greatly benefit from reporting, e.g., LPS, ESTOI, and SpkSim as generative methods can struggle with keeping faithful content. For a suitable set of metrics see the URGENT Challenge.
> >
> > All in all, we thank the authors again for their thorough explanations and are open to raise one or the other of our ratings in case the aforementioned remarks are adequately addressed in the revised paper.

---

> > > ### Author Response · Authors · 2025-11-28
> > > **Response to Reviewer DuQh**
> > >
> > > Overall, we sincerely thank you for your time and constructive suggestions! Your feedback has greatly helped us to improve the quality of our manuscript. We would be delighted to discuss if you have any further questions.

---

> ### Author Response · Authors · 2025-11-28
> **Response to Reviewer DuQh**
>
> Thank you for your advice for adding additional metrics. We further tested the STOI, ESTOI and LPS following the URGENT Challenge.
>
> The results are as follows:
>
> VoiceFixer-GSR
>
> | Model | STOI | ESTOI | LPS |
> |------------|------|--------|-------|
> | VoiceFixer | *0.892* | *0.763* | 0.761 |
> | UniverSE++ | 0.765 | 0.607 | 0.727 |
> | Resemble-Enhance | 0.697 | 0.544 | *0.764* |
> | VoiceBridge| **0.907** | **0.781** |**0.767** |
>
> DNS-with-Reverb
>
> | Model | STOI | ESTOI | LPS |
> |------------|------|--------|-------|
> | VoiceFixer | **0.785** | **0.633** | **0.745** |
> | UniverSE++ | 0.399 | 0.204 | 0.177 |
> | Resemble-Enhance | 0.101 | -0.029 | 0.339 |
> | VoiceBridge| *0.747* | *0.607* | *0.729* |
>
> We need to note that, however, intrusive metrics such as ESTOI and STOI do not necessarily reflect the generated audio quality, as pointed out in [1,2,3,4]. We will add these results in the revised version for completeness. Especially, in the GSR settings, degraded priors are highly distorted, leading to hard point-to-point recovery for generative models and failures for mapping-based models. This cause the fact that our tested STOI, ESTOI metrics are much lower than those reported in the VB-Demand benchmarks in other works such as [5].
>
> [1] Manocha, P., Jin, Z., and Finkelstein, A. Audio similarity is unreliable as a proxy for audio quality, 2022.
>
> [2] Manjunath, T. Limitations of perceptual evaluation of speech quality on voip systems.
>
> [3] Andreev, P., Alanov, A., Ivanov, O., and Vetrov, D. Hifi++: a unified framework for bandwidth extension and speech enhancement.
>
> [4] Babaev, N. et al. FINALLY: fast and universal speech enhancement with studio-like quality
>
> [5] Richter, J et al. Speech enhancement and dereverberation with diffusion-based generative models

---

### Official Review · Reviewer_sWdU · 2025-10-31

**Soundness:** 2
**Presentation:** 4
**Contribution:** 3
**Rating:** 4
**Confidence:** 4

**Summary:**

The paper presents a general speech restoration system based on latent bridge models that unifies diverse degradation-to-clean speech tasks in a single framework, achieving high-fidelity 48 kHz restoration through many innovations: energy-preserving VAE for structured latent space, a joint neural prior that fine-tunes the encoder to map degraded inputs closer to clean speech embeddings, and crucially perceptual-aware fine-tuning for human-aligned quality.

**Strengths:**

Results are strong, adequate empirical validation on multiple datasets and showing generalization (on reasonably matched domains though: only on unseen codec artifacts and from dns synthetic training to real).
Comparison with strong baselines.
Good writing and structuring.

**Weaknesses:**

A minor weaknes is that it is mainly incremental/engineering work as the core components ideas are already been proposed
(Schrodinger bridge for SE, and transformer backbone and VAE are adapted from existing architectures ).
However there is still much value in engineering and incremental work IMHO.

The model's total number of parameters is 544M in the transformer plus 156M each in the VAE encoder and decoder, totaling 856M parameters. This is substantially larger than baseline models such as Unverse++ (\~100M) or AnyEnhance (\~300M). As such, the comparisons in Table 1, for example, are not entirely fair. The better performance of the proposed method may stem from the increased model capacity rather than the novel methodology itself.

The use of UTMOS and PESQ losses calls some of the results into question. The comparisons in Table 1 are not entirely fair, at least regarding PESQ and UTMOS, as some baselines did not optimize directly for these metrics. The only way to rule this out, unfortunately, is by conducting larger-scale listening tests (as per Goodhart's law, PESQ and UTMOS become unreliable as evaluation metrics once they are used as objective functions e.g. see [1]).
To their credit, the authors discuss this issue in the Appendix and in the ablation section, and I find the discussion quite insightful. Looking at Table 15, the performance without these losses appears much weaker, which calls into question many of the results and the much of the effort and some design choices presented in the paper.
This paper contains numerous contributions and represents a substantial engineering effort, but it is sometimes difficult to extract clear scientific insights about which components are most important. I believe the ablation results in the Appendix (Table 15) are among the most significant findings, as they suggest that what truly makes the model better is the combined use of GAN and perceptual losses. As such, I think these results should be presented more prominently in the main manuscript (e.g. included in some results Tables maybe ?)

Unfortunately, in its current form, the manuscript would benefit from refocusing on its most impactful contributions before publication.

[1] de Oliveira, D., Welker, S., Richter, J., & Gerkmann, T. (2024). The PESQetarian: On the Relevance of Goodhart's Law for Speech Enhancement. Interspeech 2024

**Questions:**

You train two separate encoders: E for HQ and E^np for LQ. Have you tried using a single conditional encoder E(x, is_lq)?

---

> ### Author Response · Authors · 2025-11-21
> **Response to Reviewer sWdU, part 1**
>
> Thank you very much for the thoughtful review and for recognizing the strengths in empirical validation, generalization, and writing quality. We address your concerns point-by-point below.
>
> ***
>
> ## 1. "Mainly incremental / engineering" contribution
>
> > *"A minor weakness is that it is mainly incremental/engineering work as the core components ideas are already been proposed… However there is still much value in engineering and incremental work IMHO."*
>
> We appreciate this perspective and fully agree that we build on existing primitives (SB for SE, transformer backbones, VAE architectures). Our intent is not to claim that each individual ingredient is conceptually new, but that:
>
> 1. **Problem setting is new:**
> 	1. We target **general 48 kHz speech restoration** across *many* distortions (noise, reverb, band-limiting, clipping, etc.) with **one unified model**, rather than single-task denoising/dereverb/bandwidth extrapolation.
>
> 	2. Prior SB-based SE works operate in **data space** and are typically **task-specific**; they do not propose a unified GSR system at 48 kHz with a tractable number of inference steps.
>
> 2. **EP-VAE + Joint Neural Prior (JNP) are designed specifically for this multi-distortion SB setting:**
> 	1. EP-VAE enforces **energy-preserving, scale-equivariant** latents so that both HQ and varied LQ signals live in a structured latent space that respects waveform energy.
>
> 	2. JNP explicitly **contracts the LQ prior distributions toward the HQ manifold** in latent space, which is crucial for a single SB model to handle very heterogeneous degradations.
>
> 3. **Perceptual fine-tuning is studied in a new regime:**
> 	1. The combination of latent SB modeling with both **perceptual and GAN losses** is, to our knowledge, novel in GSR.
>
> 	2. Our ablations (Table 15) show that perceptual fine-tuning is **not a linear combination of loss terms**. We study how each component works and interacts in this regime, and propose a finetuning strategy that overcomes the common issues of finetuning on instrumental metrics.
>
>
> In the revised version, we will sharpen the contribution statement to emphasize: (1) the *setting* (general 48 kHz GSR with SB), (2) the *latent-space design* (EP-VAE + JNP for prior alignment), and (3) the *empirical insights* about perceptual objectives in this SB pipeline.
> We hope this clarifies that, while we indeed build on prior techniques, the way they are combined and analyzed is substantively new for this problem.

---

> ### Author Response · Authors · 2025-11-21
> **Response to Reviewer sWdU, part 2**
>
> ## 2. Parameter count and fairness of comparisons
>
>
> > *"Model has 856M params vs Universe++ ~100M, AnyEnhance ~300M; comparisons not entirely fair, performance may stem from capacity."*
>
> We agree that **VoiceBridge is larger in parameter count**, and we appreciate the chance to clarify.
>
> 1. **Why a larger model?**
> 	1. Our goal is a **single general-purpose 48 kHz GSR model** that covers a wide range of degradations and domains.
>
> 	2. This is inherently more demanding than a single-task SE model; we found that a moderately large transformer in latent space (544M) plus a high-fidelity encoder/decoder (156M each) is necessary for stable training and strong zero-shot behavior (e.g., Encodec artifact removal, TTS post-processing).
>
> 2. **Capacity vs. methodology: role of ablations**
> 	1. While cross-model comparisons are affected by parameter count, our **ablation studies are all conducted within the same architecture**, so they isolate the impact of **EP-VAE, JNP, and perceptual fine-tuning** independently of capacity.
>
> 3. **Contextualizing capacity**
> 	1. VoiceBridge uses only 4 inference steps in latent space for all results with a fast generation speed, while many strong baselines rely on deep stacks of convolutional/transformer blocks with **heavy calculation** in the data domain for each forward pass. Thus, **runtime cost is more comparable than raw parameter count.** A RTF comparison is reported in Appendix H, table 14, though we agree that a detailed FLOPs comparison would further clarify this.
>
>
> We agree that parameter count is an important factor and will make this much more transparent.

---

> ### Author Response · Authors · 2025-11-21
> **Response to Reviewer sWdU, part 3**
>
> ## 3. Goodhart’s law, UTMOS/PESQ losses, and scientific insight
>
>
> > *"Using UTMOS and PESQ losses calls results into question… baselines did not optimize for these metrics… Table 15 shows performance without these losses appears much weaker… results suggest what truly makes the model better is combined use of GAN and perceptual losses… this should be more prominent."*
>
> Thank you for appreciating our ablation findings. We ourselves also find the ablation study in Table 15 highly insightful. We largely agree with your concerns and appreciate the reference to *"The PESQetarian"*[1] .
>
> 1. **Risk of metric over-optimization and Goodhart’s law**
> 	1. We **do not rely solely** on PESQ/UTMOS as metrics: our evaluation includes **DNSMOS, NISQA, and, most important of all, human MOS listening tests** (Table 2). In Table 2, VoiceBridge yields **substantial gains in human MOS** on both VCTK VoiceFixer-GSR and DNS-Real, indicating that the improvements are not an artifact of overfitting PESQ/UTMOS.
>
> 	2. We indeed find experimental results supporting the findings in PESQetarian. As shown in the second-last line in Table 15, adding human feedback losses largely raises the corresponding metrics, but show significant drops on other metrics such as NISQA. Meanwhile, a severe artifact is found in listening experiments (where the high-frequency "/s/" consonant brings blurred spectrograms and a weird sound). This, we believe, is the result of training objective hacking during the finetuning process.
>
> 	3. Jointly adding a GAN objective successfully solved the above issue. Shown in Table 15, combining the losses achieves consistent improvements across all metrics. It can be interpreted that the artifacts brought by overfitting on the metric related objectives can be easily spotted by the discriminator. With the adversarial supervision from the discriminator, the model is forced to learn to increase the metrics by improving generation quality.
>
> 2. **Role of UTMOS/PESQ losses in training**
> 	1. These losses are applied only in the **final perceptual refinement stage**; the core SB + EP-VAE + JNP training is driven by reconstruction and bridge objectives.
>
> 	2. Thus, while they indeed push the model to improve specific metrics, the learned latent structure and SB dynamics are not *solely* driven by PESQ/UTMOS.
>
> 3. **Fairness to baselines that did not optimize these metrics**
> 	1. To mitigate the inclusion of PESQ/UTMOS related losses, we **emphasize human MOS results** where baselines have no "optimization advantage", and report **multiple perceptual metrics** instead of cherry-picking the ones directly optimized.
>
>
> We agree and are very grateful to hear your advice that the story should be refocused, to stress that one of the main contributions is precisely this empirical finding: **combining the GAN objectives novelly solved the problem raised by PESQetarian, and bridged the gap between direct perceptual metrics and true perceptual quality.** Making it possible for perceptual finetuning to match and surpass high-quality baselines on perceptual metrics and human MOS.
>
> - In the revised version we will:
> 	- Move the most important ablation findings (Table 15 or a condensed version) into the **main paper**, clearly highlighting the performance changes with different finetuning objectives.
>
> 	- Make the Goodhart’s law concern explicit in the main text, citing [1], and clarify how we interpret the gains and their limits.
>
> [1] de Oliveira, D., Welker, S., Richter, J., & Gerkmann, T. (2024). The PESQetarian: On the Relevance of Goodhart's Law for Speech Enhancement. Interspeech 2024

---

> ### Author Response · Authors · 2025-11-21
> **Response to Reviewer sWdU, part 4**
>
> ## 4. On using two encoders vs. a single conditional encoder
>
>
> > *"You train two separate encoders: $\mathcal{E}$ for HQ and $\mathcal{E}^\text{np}$ for LQ. Have you tried using a single conditional encoder $\mathcal{E}(x, \text{is}\text{lq}) $ ?"*
>
> This is an excellent question.
>
> - Conceptually, we chose **two encoders with shared architecture but separate weights** for the following reasons:
> 	- Keep the HQ encoder  $\mathcal{E}$ stable (In training, it defines the ground truth latent manifold). This is the most important reason, because we do not want to degrade the GT latent quality.
>
> 	- If we used a single conditional encoder, JNP training updates could potentially **pollute the HQ latent** or entangle HQ/LQ representations in undesirable ways.
>
> - We have not yet trained a full conditional encoder $\mathcal{E}(x, \text{is}\text{lq}) $ variant under the same setup, mainly due to compute constraints and our desire to ensure HQ latents remain a fixed reference. However, we agree this is a very interesting direction:
> 	- One could imagine weight-sharing between $\mathcal{E}$  and $\mathcal{E}^\text{np}$ with a small conditional modulation, or a regularizer to keep HQ latents stable while pulling LQ latents closer.
>
> 	- We will add this to the discussion and future work section, explicitly outlining why we expect two encoders to be more stable in the current scheme and how a conditional encoder might be designed as a more parameter-efficient alternative.
>
>
> ***
>
> Once again, thank you for your detailed and constructive feedback. We will:
>
> - Make model capacity and metric usage more transparent,
>
> - Move the most revealing ablations (particularly Table 15) into the main text, and
>
> - Refocus the narrative to emphasize the **empirical insights about SB + latent modeling + perceptual objectives**, which we believe are useful to the community beyond this specific architecture.

---

> > ### Comment · Reviewer_sWdU · 2025-11-26
> > **Thank for addressing my concerns**
> >
> > I increased my score to 6 after careful consideration.

---

> > > ### Author Response · Authors · 2025-11-27
> > > **Thanks to Reviewer sWdU**
> > >
> > > Thank you very much for your time and valuable suggestions. It is very helpful for us to improve the quality of the manuscript.

---

### Official Review · Reviewer_2nsd · 2025-11-01

**Soundness:** 2
**Presentation:** 2
**Contribution:** 2
**Rating:** 2
**Confidence:** 4

**Summary:**

This paper proposes to tackle the general speech restoration task from the latent space with a Schrödinger Bridge model. To improve the performance of latent diffusion models in tackling diverse distortions in audio signals, an energy-preserving variational autoencoder and a joint neural prior are proposed. Furthermore, a refinement stage is applied for perceptual enhancement. Extensive experiments are conducted to verify the effectiveness of the proposed methods on diverse benchmarks.

**Strengths:**

1. The proposed joint neural prior, which uniformly reduces the distance between each degradation prior and the target, offers a promising approach for general speech generation and holds potential for advancing related domains.

2. This paper presents novel experiments on codec-artifact removal and post-processing for TTS systems.

**Weaknesses:**

1. The motivation for performing general speech restoration in the latent space rather than the data space is not clearly stated in the paper.
Additionally, latent diffusion has been widely applied in other domains (e.g., computer vision and audio generation); its novelty in the context of speech restoration remains limited. Clearer technical distinctions or improvements over prior work should be provided.

2. Based on the experimental results, the proposed VoiceBridge does not bring promising improvements compared with other baseline models, providing insufficient evidence of the necessity of adopting latent diffusion.

3. The organization of the manuscript has minor structural issues, where the VoiceBridge training pipeline is only briefly described in the experimental setup section rather than being presented comprehensively in the methodology part.

4. The proposed perceptual-aware generation process has been a common practice applied in the speech restoration task and audio generation task, thus lacking sufficient novelty.

**Questions:**

1. Based on the experimental results, the adopted joint neural prior, which aims to uniformly reduce the distance between each degradation prior and the target, contributes to the overall improvements. However, it remains unclear to readers whether the adopted joint neural prior benefits all distortions equally or actually excels for certain distortions while underperforming for others. It would be more convincing to provide the results on each degradation to verify the proposed method.

2. The author could provide auxiliary ablation studies on performing latent diffusion from the data space to verify the effectiveness of the latent diffusion for the general speech restoration task.

3. The formatting of the references is inconsistent. I recommend unifying the citation style and ensuring accurate, case-sensitive capitalization of all titles.

---

> ### Author Response · Authors · 2025-11-21
> **Response to Reviewer 2nsd, part 1**
>
> Thank you for the careful review and for highlighting the promise of the joint neural prior and our new experiments on codec-artifact removal and TTS post-processing. Below we address your concerns point-by-point.
>
> ***
>
> ## 1. Motivation of *latent* Schrödinger Bridge for GSR
>
>
> > *"The motivation for performing general speech restoration in the latent space rather than the data space is not clearly stated"*
>
> Our goal is **general speech restoration at full-band 48 kHz across diverse distortions** (noise, reverb, bandwidth loss, clipping, etc.), using **one** model. Doing this directly in waveform / STFT / Mel space with a Schrödinger Bridge (SB) has two major issues:
>
> 1. **Computational cost and scalability.**
> 	1. At 48 kHz, even a short utterance corresponds to tens of thousands of samples. Directly performing SB in the data space takes great modeling and computational effort. Meanwhile, raw speech signals include massive redundant information at high frequencies, which can be effectively compressed by VAEs, resulting in a more compact latent space. Latent compression reduces sequence length by **more than an order of magnitude**, enabling a single 544M transformer LBM to handle all tasks.
>
> 2. **Multi-task generalization and prior alignment.**
> 	1. In data-space SB, each degradation type induces a **very different prior manifold**, making joint modeling across many distortions difficult.
>
> 	2. Our **EP-VAE** explicitly enforces **scale-equivariant, energy-preserving** behavior (Sec. 3.2, Eq. (5)), so that latent representations faithfully reflect waveform energy at multiple scales rather than at a single scale as in vanilla VAEs.
>
> 	3. The **joint neural prior (JNP)** (Sec. 3.3, Eq. (6), Fig. 2) further **contracts the distribution of LQ priors toward the HQ manifold** in latent space, improving the conditioning for a single SB model across tasks. This "prior convergence" is much harder to achieve directly on raw waveforms.
>
> To prove that latent space is indeed a superior choice for GSR, we conducted an additional ablation experiment, training three bridge models on VAE latent, waveform, and complex STFT data space respectively. For the waveform-space bridge, we adopt the diffwave[1] architecture. For the STFT space, the NeMo[2] (adopted in [3]) architecture is used. **All networks are scaled to around 540M parameters** by increasing the number of layers and hidden channels. For the VAE latent, we use VoiceBridge's configuration **without the perceptual-aware finetuning process**, for equal comparison on the modeling spaces. All models are trained for 500k steps on the same data mixture as VoiceBridge. The results on two benchmarks are as follows:
>
> On VoiceFixer-GSR:
>
> | Model          | PESQ  | SIG | BAK | OVRL | UTMOS | WV-MOS | NISQA |
> |----------------|-------|-----|-----|------|-------|--------|-------|
> | Waveform SB | 1.56 | 2.65 | 3.83 | 2.41 | 2.21 | 2.87 | 1.73 |
> | STFT SB     | 2.06 | 3.07 | 3.34 | 2.58 | 2.69 | 2.44 | 2.73 |
> | Latent SB   | **2.07** | **3.43** | **4.09** | **3.20** | **3.58** | **3.86** | **3.76** |
>
> On DNS-Real:
>
> | Model          | SIG | BAK | OVRL | UTMOS | WV-MOS | NISQA |
> |----------------|-----|-----|------|-------|--------|-------|
> | Waveform SB  | 2.64 | 2.55 | 1.94 | 1.61  | 1.55 | 1.56 |
> | STFT SB       | 2.96 | 3.57 | 2.55 | 2.27  | 2.59 | 3.39 |
> | Latent SB     | **3.39** | **4.03** | **3.12** | **3.15** | **3.69** | **4.34** |
>
> As we can see, under **equal settings** of parameter count, training data, and training iterations, latent SB far outperforms waveform and STFT space SBs with a significant advantage, providing a **clear motivation** for its use in GSR.
>
> In the revised version, we will clarify this more explicitly in **Sec. 3.1–3.3**, including:
>
> - A short paragraph contrasting data-space SB vs. latent SB for GSR,
>
> - A clearer explanation that EP-VAE + JNP are specifically designed to make **multi-distortion latent SB** feasible and stable.
>
> - An ablation on the choice of the modeling space in Appendix.
>
> [1] Kong Z, Ping W, Huang J, et al. Diffwave: A versatile diffusion model for audio synthesis.
>
> [2]  E. Harper et al. NeMo: a toolkit for conversational ai and large language models.
>
> [3] Pin-Jui Ku, Alexander H Liu, Roman Korostik, Sung-Feng Huang, Szu-Wei Fu, and Ante Jukic. Generative speech foundation model pretraining for high-quality speech extraction and restoration.

---

> ### Author Response · Authors · 2025-11-21
> **Response to Reviewer 2nsd, part 2**
>
> ## 2. Novelty relative to prior latent diffusion / latent audio work
> > *latent diffusion has been widely applied… novelty in the context of speech restoration remains limited.*
>
> We agree that latent diffusion has been widely explored, but to our knowledge **no prior work combines**:
>
> - A **tractable Schrödinger Bridge** (data-to-data generative process)
>
> - A **48 kHz general speech restoration** setting spanning many distortions and domains
>
> - With an **energy-preserving latent model** and a **joint neural prior** designed specifically to contract diverse degradation priors toward the target latent manifold.
>
>
> Existing SB-based speech works operate **in data space** and are typically **task-specific** (denoising, dereverb, bandwidth extrapolation, etc.), not a unified GSR system. Latent diffusion works for audio generally **do not tackle general speech restoration across many distortions**, nor do they introduce EP-VAE style energy-preserving constraints or JNP-like prior convergence for GSR.
> We will strengthen the related work section to explicitly position VoiceBridge with respect to data-space SB for SE (prior work), and latent diffusion / latent audio generation (different problem setting and objectives).

---

> ### Author Response · Authors · 2025-11-21
> **Response to Reviewer 2nsd, part 3**
>
> ## 3. Strength of experimental improvements & necessity of latent SB
>
>
> > *"VoiceBridge does not bring promising improvements compared with other baseline models… insufficient evidence of necessity of adopting latent diffusion."*
>
> We respectfully disagree and believe our results **do show strong and consistent gains**, especially in human listening tests and **perceptually aligned metrics**, demonstrating that Voicebridge generates clear, high-quality speech, which are most important for GSR.
>
> **(a) Human listening tests (Table 2)**
> On a subset evaluated with human MOS:
>
> - **VCTK VoiceFixer-GSR:**  VoiceBridge 4.32 vs. 3.77 (VoiceFixer), 3.97 (Resemble-Enhance), 2.91 (UniverSE++), and close to ground truth 4.43.
>
> - **DNS-Real:**  VoiceBridge 4.28 vs. 2.90–3.02 for baselines.
>
>
> VoiceBridge **substantially improves perceptual quality**, significantly outperforming other baselines in human subjective opinions.
>
> **(b) Main GSR benchmarks (Table 1)**
> On **VCTK VoiceFixer-GSR**:
>
> - **PESQ:**  2.436 (VoiceBridge) vs. 2.346 (best baseline, UniverSE++).
>
> - **UTMOS:**  4.330 vs. 3.608 (Resemble-Enhance).
>
> - **WV-MOS:**  4.246 vs. 4.224 (Resemble-Enhance).
>
> - We are best or second-best on **6 out of 7 metrics**, only slightly behind on NISQA.
>
>
> On **DNS-with-Reverb**:
>
> - **PESQ:**  1.521 (VoiceBridge) vs. 1.407 (VoiceFixer) / 1.154 (Resemble-Enhance).
>
> - **UTMOS:**  3.373 vs. 2.756 (Resemble-Enhance).
>
> - **WV-MOS:**  4.188 vs. 3.248 (Resemble-Enhance).
>
> - Again, VoiceBridge is best or second-best across all metrics.
>
>
> On **DNS-Real**:
>
> - VoiceBridge achieves the highest **SIG, BAK, OVRL, UTMOS, WV-MOS, and NISQA**, significantly surpassing all baselines.
>
>
> These improvements are especially notable in the **MOS predictors (UTMOS/WV-MOS/NISQA)** that correlate with human judgments.
>
> **(c) Zero-shot experiments on other tasks**
> As you noted in your strengths:
>
> - **Codec artifact removal (Table 4):**  VoiceBridge, though never trained on Encodec artifacts, achieves the best or near-best scores across PESQ, UTMOS, DNSMOS, and NISQA, showing strong **zero-shot robustness**.
>
> - **TTS/post-processing (Table 5):**  When applied to the outputs of recent TTS/podcast models, VoiceBridge consistently **improves both GSR and TTS metrics**, while other GSR baselines sometimes fail to refine high-fidelity generators. Notably, VoiceBridge is **the only model** that can make further enhancement to the generated audio from the conventional TTS model MoonCast.
>
> - Downstream tasks (Table 3, 7, 8, 9): VoiceBridge demonstrated consistent improvement over other general models and specialized models (SGMSE, StoRM, AudioSR, NuWave2, etc.) on Denoising, Bandwidth Extension, and Dereverberation tasks.
>
>
> We will highlight these improvements more clearly in the main text (e.g., bolding key gains in MOS and zero-shot tasks and adding a short qualitative summary in Sec. 4.2–4.4), as it seems our current framing may have under-emphasized them.

---

> ### Author Response · Authors · 2025-11-21
> **Response to Reviewer 2nsd, part 4**
>
> ## 4. Organization: training pipeline vs methodology
>
>
> > *"The VoiceBridge training pipeline is only briefly described in experimental setup rather than in the methodology part."*
>
> This is a fair comment. Currently, the four-stage pipeline (pre-train EP-VAE → joint neural prior fine-tuning → latent SB training → perceptual fine-tuning) is summarized in **Sec. 4.1**.
> In the revised version, we will **move and expand this description into Sec. 3** as an explicit **"Training Pipeline Overview" subsection**, right after introducing EP-VAE and JNP, and cross-reference it from Sec. 4.1. This should make the overall design of VoiceBridge easier to follow without jumping to the experimental section.

---

> ### Author Response · Authors · 2025-11-21
> **Response to Reviewer 2nsd, part 5**
>
> ## 5. Perceptual-aware generation: novelty and role
>
>
> > *"The proposed perceptual-aware generation process has been a common practice… thus lacking sufficient novelty."*
>
> We agree that individual components such as adversarial loss, feature matching loss, or human feedback related loss are not themselves new, and we do not intend to claim otherwise.
> Our contribution here is **how** they are integrated into the **l**atent SB-based GSR pipeline, to make comprehensive improvements over the model and prevent adversarial hacking to the MOS-predictors:
>
> - We **freeze the EP-VAE encoder** and jointly fine-tune the **LBM + decoder** with a mix of perceptual losses, preserving the carefully structured latent manifold created by EP-VAE and JNP.
>
> - The ablation study in **Table 6, Sec. 4.5 and Table 15, Appendix I** shows that this stage notably improves perceptual metrics, and giving other insights indicating that our joint finetuning process is not just a linear combination of the several losses. Specifically:
> 	- Conducting perceptual aware finetuning with the decoder kept frozen *cannot* surpass the continue-training baseline. This demonstrates that finetuning the decoder to adapt to the latent manifold of the SB's output is equally significant.
>
> 	- Adding human feedback related loss terms based *cannot* consistently improve other metrics that are not related to the used metric predictor. It even induced artifacts in the generated audio. This finding is consistent with PESQetarian[4], demonstrating the model may hack the training objective.
>
> 	- However, we solved the above issue by adding the **adversarial and feature matching terms**, which protect the model from overfitting to any special cases, **guaranteeing the model indeed learns to improve perceptual quality through the human feedback predictor**. This approach, as far as we are concerned, has never been raised in prior works.
>
> - This is particularly important for GSR because the SB framework tends to optimize posterior likelihood / reconstruction in latent space; without perceptual fine-tuning, high perceptual quality on waveforms is not guaranteed.
>
>
> We will clarify in the paper that the novelty lies in the **combination** of SB-based latent modeling with EP-VAE, JNP, and a carefully staged perceptual refinement procedure, not in inventing new perceptual losses per se.
>
> [4] de Oliveira, D., Welker, S., Richter, J., & Gerkmann, T. (2024). The PESQetarian: On the Relevance of Goodhart's Law for Speech Enhancement

---

> ### Author Response · Authors · 2025-11-21
> **Response to Reviewer 2nsd, part 6**
>
> ## 6. Responses to specific questions
>
>
> **(Q1) Does the joint neural prior benefit all distortions equally?**
> We agree that this is an important question. JNP is designed to **reduce the distance between each LQ prior and the HQ target** (Sec. 3.3, Fig. 2, Eq. (6)). As shown in **Figure 5 in Appendix G**, though **never guaranteed to be exactly "equal"**, the Joint Neural Prior indeed reduced the distance of prior distributions across all distortions to a similar scale. To further prove that the JNP indeed benefits all distortions, we add an auxiliary ablation experiment, evaluating the effect of JNP on restoring different degradations. We train two models with and without JNP respectively, and evaluate the gains on different degradation scenarios (on curated testsets with additional noise, reverberation, bandwidth limitation, and clipping on the VCTK corpus). The models are trained for 500k steps.
>
> | Noise  | PESQ | DNSMOS | WVMOS | UTMOS | NISQA |
> |--------|------|--------|-------|-------|-------|
> | w/ JNP | 1.88 | 3.17 | 4.05 | 3.69 | 4.36 |
> | w/o JNP | 1.75 | 2.95 | 4.01 | 3.65 | 4.10 |
>
> | Reverb | PESQ | DNSMOS | WVMOS | UTMOS | NISQA |
> |--------|------|--------|-------|-------|-------|
> | w/ JNP | 2.12 | 3.24 | 4.11 | 3.77 | 4.60 |
> | w/o JNP | 2.01 | 3.13 | 4.10 | 3.72 | 4.43 |
>
> | BWE | PESQ | DNSMOS | WVMOS | UTMOS | NISQA |
> |--------|------|--------|-------|-------|-------|
> | w/ JNP | 2.70 | 3.30 | 4.32 | 3.97 | 4.69 |
> | w/o JNP | 2.51 | 3.19 | 4.26 | 3.95 | 4.50 |
>
> | CLIP | PESQ | DNSMOS | WVMOS | UTMOS | NISQA |
> |--------|------|--------|-------|-------|-------|
> | w/ JNP | 3.26 | 3.25 | 4.31 | 4.01 | 4.66 |
> | w/o JNP | 2.95 | 3.16 | 4.20 | 3.91 | 4.63 |
>
> The results demonstrate consistent improvement across all dereverberation types, indicating that JNP benefit all distortions simultaneously.
>
>
> **(Q2) Ablation: latent SB vs. data-space SB**
> We sincerely agree that adding such an auxiliary ablation study on the design space of SB modeling is essential for the choice of VAE latent space. We provide the ablation result in Section 1 (a) of this response. We will also add this ablation study to the appendix in the revised version.
>
>
> **(Q3) Reference formatting**
> Thank you for pointing it out. We acknowledge the inconsistency in reference formatting (capitalization, styles). We will clean and unify the citation style and ensure correct case-sensitive titles in the revised version.
>
> ***
>
> Again, thank you for your detailed comments. We will revise the paper to more clearly articulate:
>
> 1. The **motivation and advantages** of latent SB with EP-VAE and JNP for general 48 kHz GSR,
>
> 2. The **strength of our perceptual and zero-shot improvements** over strong baselines, and
>
> 3. The **overall pipeline and structure** of VoiceBridge, alongside additional ablations and per-degradation analyses.

---

> ### Author Response · Authors · 2025-11-28
> **Follow-up on responses to Reviewer 2nsd**
>
> Dear Reviewer 2nsd
>
> We hope this message finds you well. We are writing to follow up on my recent responses to your comments on our manuscript. Could you please let me know whether our answers have addressed your concerns, or if there are any remaining issues that you would like me to clarify or revise further? Your feedback is very important for improving the manuscript, and I would greatly appreciate any additional guidance you might have. Thank you!

---

### Author Response · Authors · 2025-11-30
**Overall Response**

We thank the reviewers for their thoughtful feedback. In the revised manuscript, we have made the following main changes and additions:

1. **Clarified motivation and contributions.**

   * We sharpen the contribution statement around three pillars: (i) an energy-preserving VAE (EP-VAE) that couples waveform energy and latent magnitude, (ii) a joint neural prior (JNP) that contracts diverse LQ priors toward a shared HQ manifold, and (iii) a perceptually aware fine-tuning regime that combines metric-based and GAN-based objectives while mitigating Goodhart-style metric hacking.
   * In Sec. 3, we now more explicitly motivate performing SB in **latent space** instead of waveform/STFT space at 48 kHz, emphasizing scalability and multi-distortion generalization.
   * We add a new ablation in Appendix I, studying comparing SB in waveform, STFT, and latent spaces under matched settings, showing clear advantages of latent SB.

2. **Added a “Training Pipeline Overview” in the method section.**

   * We introduced a new subsection in Sec. 3 that clearly describes the four-stage pipeline: EP-VAE pre-training, JNP alignment for degraded inputs, latent SB training, and joint perceptual fine-tuning of the LBM and decoder.
   * Sec. 4.1 now focuses on detailed experimental settings, improving the overall organization.

3. **New and reorganized ablations.**
  More ablation studies were added in Appendix I:
   * We added an ablation comparing SB in waveform space, STFT space, and latent space under matched parameter counts and training settings, showing clear advantages of latent SB.
   * We included per-degradation JNP ablations (noise, reverb, BWE, clipping), demonstrating that JNP improves all distortions rather than a subset.
   * We added a comparison between a “KL-only” noise-aware prior (Fang et al.-style) and our full JNP objective, supporting the benefit of multi-measure alignment.
   * We report VAE-only enhancement $ \hat{x}_0 = D(E\_{\text{np}}(x\_1)) $ versus the full SB model, including per-degradation results, clarifying when the VAE already helps and where the bridge provides additional gains.
   * The most informative perceptual fine-tuning ablations (previously only in the appendix) are now summarized in the main text, highlighting how metric-only losses can harm other metrics and listening quality, and how adding adversarial + feature-matching terms resolves this.

4. **Perceptual objectives and Goodhart’s law.**

   * We explicitly discuss the risk of optimizing PESQ/UTMOS as objectives (Goodhart’s law), cite “The PESQetarian,” and connect it to our ablation findings in Section 4.5.
   * The “human feedback” loss $L_{\text{hf}}$ has been renamed to a perceptual loss (e.g., $L_{\text{perc}}$) and is clearly described as using frozen PESQ/UTMOS predictors rather than online human feedback or RLHF.

5. **Baselines and cross-paper comparisons.**

   * We added Miipher as an additional baseline on our main GSR benchmarks (e.g., VoiceFixer-GSR, DNS-Real).
   * We included cross-paper comparisons with SpeechFlow and Ku et al. by downsampling VoiceBridge outputs to 16 kHz on VB-Demand, WSJ0-CHiME3, and WSJ0-BWE, and discussed the differences in sampling rate, pretraining scale, and task-specific fine-tuning in Appendix F.
   * Using released evaluation samples, we compared VoiceBridge against FINALLY and HiFi-GAN-2 on their own test material, reporting DNSMOS/UTMOS/WV-MOS and discussing the results in the context of foundation-model pretraining vs public-data training and generative vs point-to-point modeling in Appendix F.

6. **Additional clarifications and formatting.**

   * We cleaned up and unified the bibliography formatting (capitalization and citation style) and updated the related-work discussion to more explicitly position our EP-VAE and JNP with respect to Fang et al., Karras et al., Richter et al., SpeechFlow, FINALLY, HiFi-GAN-2, and other recent SE/GSR methods.

We hope these changes address the main concerns about motivation, novelty, empirical evidence, and clarity, and we thank the reviewers again for their very helpful suggestions.

---

### Author Response · Authors · 2025-11-30
**Summary of Post-Rebuttal Reviewer Feedback and Score Updates**

Dear Area Chair,

We hope this message finds you well. We understand the additional burden placed on the program committee due to the recent OpenReview issues (including the Nov. 27 identity leak) and sincerely appreciate your efforts in continuing to uphold the integrity of the ICLR review process. As you have been newly assigned to our submission, we would like to provide a brief and factual summary of the discussion and review status of our paper **prior to the Nov. 27 identity leak**, for your consideration.

* The original scores were **6, 4, 4, 2**.
* After reading our rebuttal, **Reviewer sWdU explicitly updated their score to 6**, acknowledging that their main concerns had been addressed.
* **Reviewer DuQh indicated in their comments that they would be willing to increase their score by 2 points or more** if we implemented the requested changes. We have followed their suggestions and made all the requested revisions in the updated manuscript, including clarified novelties, added ablations, new baselines, detailed discussions, and renaming of certain terms to avoid confusion.
* **Reviewer 2nsd who gave a score of 2 unfortunately did not participate in the rebuttal phase at all** and did not respond to our clarifications. Nonetheless, in our rebuttal we responded point-by-point to their concerns and believe we have effectively addressed the main issues they raised.
* **All the above discussions happen before the Nov.27 identity leak issue**.

We have invested substantial effort into a thorough and constructive rebuttal, and for the two reviewers who engaged with it, our response appears to have been effective and would have led to higher scores if the system had recorded them correctly. We kindly ask that you take our rebuttal and the reviewers’ updated qualitative feedback into full consideration when forming your recommendation, even if the numerical scores cannot be changed in the system.

Thank you very much for your time and consideration.

Best regards

---

### Meta-Review · Area_Chair_aDqS · 2026-01-02

**Summary:**

The main concerns are related to  lack of sufficient novelty, and fairness of the experimental comparisons.

**Reviewer Concerns:**

The authors have addressed some of the comments, for instance: (i) VoiceBridge does not bring promising improvements compared with other baseline models, and (ii) the novelty of the joint neural prior versus prior “noise-aware” VAEs.

Nonetheless, several key concerns remain outstanding, for example:
(i) Lack of sufficient novelty — the authors acknowledge this and state that the novelty lies in how the different components are combined, which highlights the incremental nature of the proposed contributions.
(ii) Fairness of the comparison with smaller baseline systems. The authors' answer to this point  is not fully satisfactory.
(iii) EP-VAE is not fundamentally different from what has been proposed.

**Reviewer Scores:**

Reviewer 2nsd might have kept their score unaltered.

Reviewer sWdU has substantially increased their score without clarifying which parts of their comments were resolved and which were not, making the logic behind the decision somewhat unclear.

Reviewer DuQh might have kept their score, since some of their comments were addressed while others were not properly sorted out.

Reviewer mn36 might have increased their score by +1.0.

---

### Decision · Program_Chairs · 2026-01-26

Reject